# Z-Erase: Enabling Concept Erasure in Single-Stream Diffusion Transformers

**Nanxiang Jiang** [1]  **Zhaoxin Fan** [1]  **Baisen Wang** [1]  **Daiheng Gao** [2]  **Junhang Cheng** [1]  **Jifeng Guo** [1]  **Yalan Qin** [3]  **Yeying Jin** [4]  **Hongwei Zheng** [5]  **Faguo Wu** [1]  **Wenjun Wu** [1]

## Abstract

Concept erasure is a vital safety mechanism for removing unwanted concepts from text-to-image (T2I) models. While extensively studied in U-Net and dual-stream architectures (*e.g.*, Flux), this task remains under-explored in the recent emerging paradigm of single-stream diffusion transformers (*e.g.*, Z-Image). In this new paradigm, text and image tokens are processed as a single unified sequence via shared parameters. Consequently, directly applying prior erasure methods typically leads to generation collapse. To bridge this gap, we introduce **Z-Erase**, the first concept erasure method tailored for single-stream T2I models. To guarantee stable image generation, Z-Erase first proposes a *Stream Disentangled Concept Erasure Framework* that decouples updates and enables existing methods on single-stream models. Subsequently, within this framework, we introduce *Lagrangian-Guided Adaptive Erasure Modulation*, a constrained algorithm that further balances the sensitive erasure-preservation trade-off. Moreover, we provide a rigorous convergence analysis proving that Z-Erase can converge to a Pareto stationary point. Experiments demonstrate that Z-Erase successfully overcomes the generation collapse issue, achieving state-of-the-art performance across a wide range of tasks. Code is available at: `https://github.com/nxjiang-jnx/Z-Erase`.

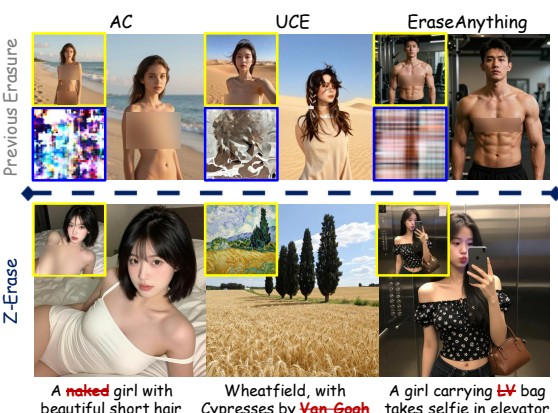

*Figure 1.* We introduce **Z-Erase**, a concept erasure method tailored for single-stream models. *First row*: Prior methods adapted via our *Stream Disentangled Concept Erasure Framework* are tested with the "nudity" concept, showing under-erasure (AC, EraseAnything) or severe artifacts (UCE). Blue boxes reveal the generation collapse caused by naive fine-tuning without our adaptation. *Second row*: Z-Erase effectively removes target concepts while preserving quality. Original outputs are in yellow boxes. Sensitive content pixelated.

## 1. Introduction

The landscape of text-to-image (T2I) generation has evolved at breakneck speed, rapidly transitioning from the foundational U-Net (Rombach et al., 2022) architectures to the scalable era of Diffusion Transformers (Peebles & Xie, 2023). Recently, this evolution has reached a new pinnacle with the emergence of single-stream diffusion transformers, a unified framework exemplified by state-of-the-art models like Z-Image (Team, 2025) and HunyuanImage-3.0 (Cao et al., 2025). Unlike traditional dual-stream models (*e.g.,* Flux (Labs et al., 2025)) that process text and pixels separately, this new paradigm fundamentally treats visual and textual data as a single, unified sequence handled by a monolithic transformer backbone (Team, 2025). The efficiency of this architectural grand unification is remarkable: Z-Image Turbo, for instance, generates stunningly photorealistic imagery with only 6B parameters, signaling that the recently emerging pure single-stream framework is not merely an architectural variant, but the blueprint shaping the next generation of foundational models.

However, this generative prowess comes with significant risks. If the training data is not carefully filtered, founda-

---

[1]Beijing Advanced Innovation Center for Future Blockchain and Privacy Computing, School of Artificial Intelligence, Beihang University. [2]University of Science and Technology of China [3]Shanghai University [4]Tencent [5]Beijing Academy of Blockchain and Edge Computing. Correspondence to: Zhaoxin Fan <zhaoxinf@buaa.edu.cn>, Faguo Wu <faguo@buaa.edu.cn>.

tional models can effortlessly absorb and reproduce copyrighted entities, Not-Safe-For-Work (NSFW) material, and biased imagery (Barez et al., 2025). As these models gain broader adoption, the societal risks of such unrestricted generation become acute. In response, concept erasure has emerged as a vital safety alignment strategy (Kumari et al., 2023; Gandikota et al., 2023). Instead of re-training the entire model, concept erasure selectively removes undesired target concepts while maintaining the model's utility and performance on normal content.

While concept erasure is well-established for Stable Diffusion (SD) (Rombach et al., 2022) and Flux (Labs et al., 2025) series, we find that directly transferring these methods to pure single-stream models leads to *consistent generation collapse*, as shown in Fig. 1 (blue boxes). Through careful analysis, we trace the failure to the unified single-stream architecture, as detailed in Section 3. Unlike previous models that isolate text-image interactions within separate modules, single-stream transformers fuse both modalities into a unified self-attention mechanism with shared weights (Team, 2025). Consequently, optimizing shared parameters to suppress text concepts will inevitably damage visual backbone's synthesis capability, leading to catastrophic noise outputs.

To tackle this issue, we introduce **Z-Erase**, a concept erasure method tailored for single-stream models. First, to enable existing erasure methods on single-stream models, we propose **Stream Disentangled Concept Erasure Framework**, a structural intervention that decouples parameter updates in single-stream models. By freezing the visual processing pathway while allowing low-rank adaptations (LoRAs) (Hu et al., 2021) only on textual hidden states, we create a safe optimization subspace that protects the image generation backbone. Within this subspace, prior erasure methods can operate on single-stream models.

However, even with this structural fix, we find that the unified attention space of single-stream models remains highly sensitive, making it difficult to balance erasure of target concept against preservation of unrelated content, as shown in Fig. 1 (first row). To address this, we further introduce a **Lagrangian-Guided Adaptive Erasure Modulation** algorithm, which solves this trade-off as a constrained optimization problem by dynamically maximizing erasure only when the preservation loss remains within a strict tolerance. In addition to the algorithm design, we also provide a rigorous theoretical analysis proving that our algorithm can converge to a Pareto stationary point (Yu et al., 2020). Experiments demonstrate that Z-Erase successfully solves this optimization dilemma, ensuring robust ethical safety with minimal and controllable degradation in utility.

To the best of our knowledge, Z-Erase is the first effective concept erasure method tailored for the emerging single-stream T2I paradigm. Our main contributions are:

- **Single-stream attention localization**. We identify that generation collapse in single-stream models stems from shared projection weights. Beyond this, we further find that attention maps allow precise token-level localization, enabling selective erasure of targeted content.

- **Stream disentangled concept erasure framework**. We propose a structural intervention for single-stream models. By injecting learnable adaptations exclusively into textual hidden states while freezing the visual backbone, we construct a safe optimization subspace that enables existing erasure methods to work on single-stream models.

- **Lagrangian-Guided Adaptive Erasure Modulation**. To resolve the sensitive trade-off between erasure and preservation, we introduce a dynamic algorithm building upon this framework that maximizes erasure within a strict utility tolerance, and provide a rigorous convergence guarantee to a Pareto stationary point.

## 2. Related Work

**Generative diffusion and flow transformers.** The evolution of generative models has progressed rapidly, moving from the foundational U-Net (Ronneberger et al., 2015) architectures of DDPM (Ho et al., 2020) and DDIM (Song et al., 2020) to the scalable era of Diffusion Transformers (DiTs) (Peebles & Xie, 2023) and flow matching (Lipman et al., 2022). Dominant T2I models (*e.g.*, Flux (Labs et al., 2025), SD3 (Esser et al., 2024), LongCat (Team et al., 2025)) establish the standard by using dual-stream architectures, which process text and visual data separately before fusing them for image generation. Recently, however, Z-Image (Team, 2025) has broken this standard by introducing a pure single-stream paradigm. By concatenating text and image tokens at the sequence level to serve as a unified input stream, Z-Image achieves strong performance with remarkable efficiency: ranking among the top open-source models in recent ELO leaderboards with only 6B parameters. Recognizing this shift, we conduct our main experiments on Z-Image to study concept erasure and safety in the next generation of foundational models.

**Concept erasure.** As generative models improve, safety concerns become increasingly important. Concept erasure offers a practical way to mitigate risks like NSFW content and copyright infringement, serving as a more efficient alternative to pre-training filtering (Rombach et al., 2022) or post-generation filtering (Rando et al., 2022). The field matures rapidly on noise-prediction U-Nets. For example, AC (Kumari et al., 2023) and ESD (Gandikota et al., 2023) fine-tune cross-attention modules by aligning predicted noise via MSE. FMN (Zhang et al., 2023) minimizes attention activations associated with target concept text tokens. UCE (Gandikota et al., 2024) solves a closed-form

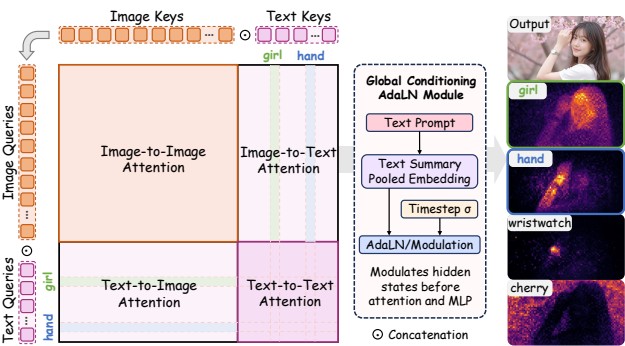

*Figure 2.* **Single-stream attention analysis.** Given the prompt *"A girl with a wristwatch on her hand amid cherry blossoms"*, the attention maps reveal distinct localized responses for text tokens.

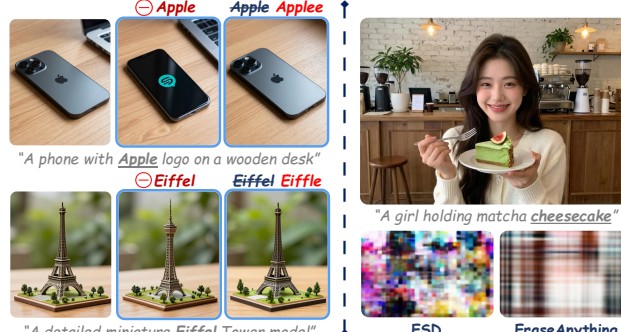

*Figure 3. Left:* **Attention erasure is brittle.** Concept can be erased by zeroing its attention columns $\mathbf{A}_{attn}[:,:,idx] = 0$ once the target token is localized from the prompt. However, this trick shows poor robustness to real-world prompt variations. *Right:* **Naive fine-tuning collapses.** Directly fine-tuning shared projections in single-stream models collapses generation with noisy outputs.

optimization of the text projection matrix. MACE (Lu et al., 2024) scales the capability to handle multiple concepts simultaneously. Subsequent works (Bui et al., 2024; Zhou et al., 2025; Kim & Qi, 2025) leverage LoRA and adversarial training to improve erasure efficiency while balancing irrelevant concept preservation. Recently works like EraseAnything (Gao et al., 2025a) and MCE (Zhang et al., 2025) successfully adapt concept erasure to Flux, tuning the separate prompt branches within its dual-stream structure. However, these methods cannot be directly applied to pure single-stream models like Z-Image, which differ fundamentally in architecture and generation dynamics. In this work, we identify the critical obstacles in adapting concept erasure to this unified paradigm and propose the first effective erasure method designed for single-stream transformers.

## 3. Obstacles in Migrating Concept Erasure to Single-Stream Models

In this section, we analyze why concept erasure methods designed for SD and dual-stream models (*e.g.*, Flux) fail when applied directly to single-stream architectures. Specifically, we highlight several fundamental obstacles, including the absence of explicit cross-attention mechanisms, the strong coupling between textual and visual representations induced by shared self-attention, and the limited robustness of naive localization-based strategies.

**Absence of explicit cross-attention:** The first key challenge is single-stream models like Z-Image lack explicit cross-attention layers (see Appendix A for detailed architecture). As a result, methods such as ESD, UCE, and MACE, which originally optimize cross-attention parameters should be redesigned to adapt the new architecture.

This observation naturally raises a question: *Although explicit cross-attention does not exist, does single-stream transformers exhibit a similar pattern between text and image tokens?* Motivated by this possibility, we conduct an in-depth examination of the neurons and features within the

single-stream architecture.

**Single-stream attention:** We ultimately find that the similar pattern of text–image interactions is encoded through self-attention. Specifically, in modern single-stream models (like Z-Image), visual and textual information are not processed by separate encoders but are instead treated as a single unified sequence. Let $\mathbf{H} \in \mathbb{R}^{(n_I+n_T)\times d}$ denote the input hidden states. The sequence concatenation is defined as:

$$\mathbf{H} = \texttt{concat}(\mathbf{H}_{img}, \mathbf{H}_{txt}) \in \mathbb{R}^{(n_I+n_T)\times d}, \quad (1)$$

where $\mathbf{H}_{img} \in \mathbb{R}^{n_I \times d}$ and $\mathbf{H}_{txt} \in \mathbb{R}^{n_T \times d}$. Unlike dual-stream architectures, which employ separate projection matrices for vision and language, single-stream models use shared weights $\mathbf{W}_Q, \mathbf{W}_K, \mathbf{W}_V \in \mathbb{R}^{d \times d_k}$ for both modalities. The query, key, and value matrices are computed as:

$$\mathbf{Q} = \mathbf{H}\mathbf{W}_Q = \begin{bmatrix} \mathbf{H}_{img}\mathbf{W}_Q \\ \mathbf{H}_{txt}\mathbf{W}_Q \end{bmatrix} = \begin{bmatrix} \mathbf{Q}_{img} \\ \mathbf{Q}_{txt} \end{bmatrix},$$
$$\mathbf{K} = \mathbf{H}\mathbf{W}_K = \begin{bmatrix} \mathbf{H}_{img}\mathbf{W}_K \\ \mathbf{H}_{txt}\mathbf{W}_K \end{bmatrix} = \begin{bmatrix} \mathbf{K}_{img} \\ \mathbf{K}_{txt} \end{bmatrix}, \quad (2)$$
$$\mathbf{V} = \mathbf{H}\mathbf{W}_V = \begin{bmatrix} \mathbf{H}_{img}\mathbf{W}_V \\ \mathbf{H}_{txt}\mathbf{W}_V \end{bmatrix} = \begin{bmatrix} \mathbf{V}_{img} \\ \mathbf{V}_{txt} \end{bmatrix}.$$

The self-attention operation $\mathbf{A}_{attn} = \texttt{Softmax}(\frac{\mathbf{Q}\mathbf{K}^\top}{\sqrt{d_k}})$ then yields a $2 \times 2$ block matrix representing intra- and inter-modality interactions:

$$\mathbf{A}_{attn} = \texttt{Softmax}\left( \frac{1}{\sqrt{d_k}} \begin{bmatrix} \underbrace{\mathbf{Q}_{img}\mathbf{K}_{img}^\top}_{\mathbf{A}_{I \leftarrow I}} & \underbrace{\mathbf{Q}_{img}\mathbf{K}_{txt}^\top}_{\mathbf{A}_{I \leftarrow T}} \\ \underbrace{\mathbf{Q}_{txt}\mathbf{K}_{img}^\top}_{\mathbf{A}_{T \leftarrow I}} & \underbrace{\mathbf{Q}_{txt}\mathbf{K}_{txt}^\top}_{\mathbf{A}_{T \leftarrow T}} \end{bmatrix} \right). \quad (3)$$

Here, the cross-terms $\mathbf{A}_{I \leftarrow T}$ and $\mathbf{A}_{T \leftarrow I}$ are intrinsic to the self-attention mechanism, creating *a tight coupling between image generation and text conditioning*, as shown in Fig. 2.

**Attention intervention is effective yet brittle:** The above analysis suggests that the interaction between text and image is directly formed within $\mathbf{A}_{attn}$: if we locate the target token in the prompt and zero out its attention column, the concept can be removed rather easily. However, this simple trick breaks once the prompt is even slightly altered—adding prefixes/suffixes (**Apple** → **Applee**) or intentional misspellings (**Eiffel** → **Eiffle**) is enough to bypass the erasure and still generate the concept, as shown in Fig. 3 (left). In other words, *deleting a single attention column is brittle and not robust to real-world prompt variation*, which motivates approaches that modify the model itself through fine-tuning.

**Naive fine-tuning fails:** However, we find that directly applying the fine-tuning concept erasure methods such as ESD (for SD v1.5) or EraseAnything (for Flux) on single-stream models consistently causes generation collapse, as shown in Fig. 3 (right). The failure stems from the shared projection weights $\mathbf{W}_Q, \mathbf{W}_K, \mathbf{W}_V$, which jointly serve text conditioning and image synthesis. Therefore, suppressing a textual concept will inevitably perturb the image token pathway through shared self-attention, making controlled concept erasure infeasible. These observations indicate that single-stream models require a dedicated erasure framework that respects this architectural coupling.

## 4. Method

**Overview.** To solve the obstacles identified above, we present Z-Erase. First, we introduce the *Stream Disentangled Concept Erasure Framework* for single-stream models. This structural mechanism provides a safe subspace for our erasure and preservation losses, and enables previous erasure methods to function effectively in the single-stream setting. Second, to handle the delicate trade-off between erasure and preservation, we propose a *Lagrangian-Guided Adaptive Erasure Modulation* algorithm based on this framework. Instead of using fixed weights, this novel algorithm dynamically adjusts erasure strength to strictly guarantee utility. Together, these two key parts make stable and effective erasure possible on single-stream models. Next, we introduce each in detail.

### 4.1. Stream Disentangled Concept Erasure Framework

As analyzed in Section 3, the core challenge in single-stream models is the strict coupling of text and image processing: directly fine-tuning shared weights to erase a text concept will unavoidably distort the visual signal, causing generation collapse. To resolve this architectural bottleneck, we propose the *Stream Disentangled Concept Erasure Framework*, which is designed to structurally isolate the parameter update trajectory.

Specifically, we first control which modality receives up-

dates. Let $\mathbf{S}_T = \mathtt{diag}(\mathbf{0}_{n_I}, \mathbf{I}_{n_T}) \in \mathbb{R}^{(n_I+n_T)\times(n_I+n_T)}$ be a token-wise selection operator that acts as a binary gate. This operator assigns zeros to image tokens and ones to text tokens, creating a physical barrier that restricts low-rank updates $\Delta\mathbf{W}$ exclusively to textual hidden states while bypassing the visual stream.

Formally, we apply this gating to the linear projection layers. For weights $\mathbf{W} \in \{\mathbf{W}_Q, \mathbf{W}_K, \mathbf{W}_V\}$, the modified forward pass is defined as $\mathbf{H}' = \mathbf{H}\mathbf{W} + \mathbf{S}_T\mathbf{H}(\Delta\mathbf{W})$. By factoring the hidden states $\mathbf{H}$ into visual and textual segments, this yields a clean separation in the adaptation dynamics:

$$
\begin{bmatrix} \mathbf{H}'_{img} \\ \mathbf{H}'_{txt} \end{bmatrix} = \begin{bmatrix} \mathbf{H}_{img} \\ \mathbf{H}_{txt} \end{bmatrix} \begin{bmatrix} \underbrace{\mathbf{W}}_{\text{Frozen}} & \mathbf{O} \\ \mathbf{O} & \underbrace{\mathbf{W}+\Delta\mathbf{W}}_{\text{Trainable, Concept Erasure}} \end{bmatrix}. \quad (4)
$$

This design constructs a safe optimization subspace for concept erasure in single-stream models. The key idea is to restrict gradient updates to the textual hidden states $\mathbf{H}_{txt}$ while keeping the pixel-generation backbone $\mathbf{H}_{img}$ frozen. This isolation is critical, because the projection weights jointly govern both semantic understanding and image synthesis. Without such decoupling, erasure gradients inevitably perturb the shared attention pathways and cause the generation collapse observed in naive fine-tuning. With this subspace in place, previous erasure methods become operational again in the unified single-stream architecture.

Within this safe subspace, we implement concept erasure as an adversarial fine-tuning process. Our goal is to suppress a target concept set $\mathcal{D}_{er}$, while anchoring the model on an unrelated concept set $\mathcal{D}_{pr}$ to ensure general utility remains unaffected. The two objects are defined as:

**Erasure objective.** Motivated by the negative guidance idea introduced by ESD (Gandikota et al., 2023), we start from direct trajectory intervention in flow matching. The first sub-loss in our Z-Erase aims to suppress the visual manifestation of the target concept during the denoising trajectory. Specifically, the erase loss is defined as:

$$
\mathcal{L}_{erase} = \mathbb{E}_{x_t,t,c_{er}\sim\mathcal{D}_{er}}\Big\| v_{\theta+\Delta\theta}(x_t, c_{er}, t) -
$$
$$
\big[v_\theta(x_t, \varnothing, t) - \eta\big(v_\theta(x_t, c_{er}, t) - v_\theta(x_t, \varnothing, t)\big)\big]\Big\|_2^2. \quad (5)
$$

Here, $\eta$ is the negative guidance factor and directly influences how aggressively the concept is erased. $\theta$ denotes the pretrained model parameters and $\Delta\theta$ is the learnable LoRA weights for erasure. $x_t$ denotes the latent at timestep $t$, starting from random noise $x_T$ where $T$ is the total timesteps. $v_\theta(x_t, \varnothing, t)$ corresponds to the unconditional velocity prediction (with empty prompt), while $c_{er} \in \mathcal{D}_{er}$ identifies the concept to erase. In flow matching, $v$ represents the velocity field guiding the trajectory between distributions, which makes it a natural site for concept-level intervention.

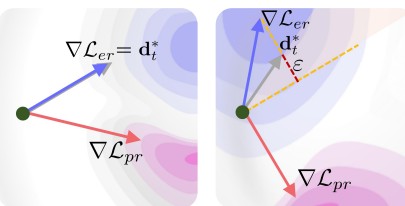

*Figure 4.* **Dynamic balancing of erasure and preservation gradients.** *Left:* When $\nabla\mathcal{L}_{er}$ does not conflict with $\nabla\mathcal{L}_{pr}$, the erasure gradient itself is the update direction. *Right:* When the gradients conflict, the update direction $\mathbf{d}_t^*$ is obtained by steering $\nabla\mathcal{L}_{er}$ toward $\nabla\mathcal{L}_{pr}$ until the preservation constraint $\varepsilon$ is satisfied.

Furthermore, building on the *single-stream attention* insights discussed in Section 3, we further reduce the influence of the erased concept by suppressing the attention assigned to its localized keywords:

$$\mathcal{L}_{attn} = \sum_{idx=start}^{end} \mathbf{A}_{attn}[:,:,idx], \qquad (6)$$

To avoid overfitting to fixed token positions, we also randomly shuffle the word order during training, keeping the index range dynamic while preserving the original meaning.

Together, these two terms form the erasure objective:

$$\mathcal{L}_{er} = \mathcal{L}_{erase} + \mathcal{L}_{attn}. \qquad (7)$$

**Preservation objective.** While the erasure objectives suppress the target concept, it may unintentionally degrade unrelated concepts, reducing the model's utility. For example, given the prompt *"A nude girl"*, our goal is to erase $c_{er}$ *"nude"* while ensuring the model can still generate images of an unrelated concept $c_{pr} \in \mathcal{D}_{pr}$ normally, *e.g.,* girl. To preserve usability, we constrain the shift in the generation process for concepts that should remain untouched:

$$\mathcal{L}_{pr} = \mathbb{E}_{x_t,t}\Big\|v_{\theta+\Delta\theta}(x_t, \varnothing, t) - v_\theta(x_t, \varnothing, t)\Big\|_2^2 +$$
$$\mathbb{E}_{x_t,t,\,c_{pr}\sim\mathcal{D}_{pr}}\Big\|v_{\theta+\Delta\theta}(x_t, c_{pr}, t) - v_\theta(x_t, c_{pr}, t)\Big\|_2^2. \qquad (8)$$

Finally, combining the erasure term, attention suppression, and preservation constraint completes the set of learning objectives for Z-Erase.

## 4.2. Lagrangian-Guided Adaptive Erasure Modulation

With our erasure and preservation objectives in place, the remaining challenge is how to optimize them together. Existing methods typically adopt linear scalarization (*e.g.,* MACE) or multi-objective optimization (*e.g.,* EraseAnything). However, simply combining erasure and preservation objectives that have inherent conflict fails to find a balanced solution, as proved in multitask learning literature (Hu et al., 2024). This failure is exacerbated in single-stream models like Z-Image, where the gradients for erasing

**Algorithm 1** Training Procedure of Z-Erase

**Input:** Pretrained model $\theta_0$, erasure concept set $\mathcal{D}_{er}$, preservation concept set $\mathcal{D}_{pr}$, total iteration steps $M$.
**Hyperparameters:** Learning rates $\alpha$ (for model), $\beta$ (for $\lambda$), tolerance $\varepsilon$, initial $\lambda_0 = 0$.
**for** iteration $t = 1$ **to** $M$ **do**
  ❶ Calculate preservation loss change:
    $\tilde{g}_t = \frac{1}{\alpha}(\mathcal{L}_{pr}(\theta_{t-1}) - \mathcal{L}_{pr}(\theta_t)) + \varepsilon$.
  ❷ Update constraint weight: $\lambda_{t+1} \leftarrow \lambda_t - \beta\tilde{g}_t$.
  ❸ Sample batches $c_{er} \sim \mathcal{D}_{er}$ and $c_{pr} \sim \mathcal{D}_{pr}$.
  ❹ Compute losses $\mathcal{L}_{er}$ and $\mathcal{L}_{pr}$ with Eq. 5 to 8.
  ❺ Compute total balanced objective:
    $\mathcal{L}_{total} = \mathcal{L}_{er} + \lambda_{t+1}\mathcal{L}_{pr}$
  ❻ Update LoRA weights:
    $\Delta\theta_{t+1} \leftarrow \Delta\theta_t - \alpha\nabla_{\Delta\theta}\mathcal{L}_{total}$
**end for**
**Output:** Final erased model $\theta' = \theta_0 + \mathbf{S}_T(\Delta\theta_M)$

a concept often clash violently with the gradients needed to preserve image quality. Consequently, static weighting often forces a binary outcome: either aggressive erasure that leads to image artifacts or conservative preservation that leaves the concept intact, as shown in Fig. 5.

To resolve this dilemma, we further introduce a *Lagrangian-Guided Adaptive Erasure Modulation* algorithm. Unlike static weighting that struggles to find a stable balance, we treat erasure and preservation as a dynamic constrained optimization problem. The goal is to converge to a controllable point on the Pareto front (Yu et al., 2020), where erasure cannot be further improved without sacrificing preservation. Specifically, we increase erasure under a preservation tolerance. At training step $t$ with parameters $\theta_t$, we solve for an update direction $\mathbf{d}_t$ that maximizes the descent of the erasure objective $\mathcal{L}_{er}$ while constraining the increase of $\mathcal{L}_{pr}$ to remain within a small tolerance $\varepsilon$:

$$\max_{\mathbf{d}_t} \nabla\mathcal{L}_{er}(\theta_t)\cdot\mathbf{d}_t - \frac{1}{2}\|\mathbf{d}_t\|^2, \text{ s.t.} \nabla\mathcal{L}_{pr}(\theta_t)\cdot\mathbf{d}_t \geq -\varepsilon, \quad (9)$$

where $\|\mathbf{d}_t\|^2$ is the regularization to avoid unbounded solutions. By introducing a Lagrange multiplier $\lambda_t \geq 0$, we can rewrite this as a dual problem:

$$\min_{\lambda_t \geq 0} \mathcal{L}_t(\lambda_t) = \frac{1}{2}\big\|\nabla\mathcal{L}_{er}(\theta_t) + \lambda_t\nabla\mathcal{L}_{pr}(\theta_t)\big\|^2 + \lambda_t\varepsilon. \quad (10)$$

This formulation shifts the optimization into finding a dynamic dual weight $\lambda_t$ that satisfies the constraint. By determining the optimal $\lambda_t$ and minimizing the Lagrangian *w.r.t.* $\mathbf{d}_t$, we derive a closed-form solution $\mathbf{d}_t^*$ that naturally takes the form of *gradient surgery*:

$$\mathbf{d}_t^* = \begin{cases} \nabla\mathcal{L}_{er}(\theta_t) + \lambda_t^*\nabla\mathcal{L}_{pr}(\theta_t), & \text{if conflict exists}(\lambda_t^* > 0) \\ \nabla\mathcal{L}_{er}(\theta_t), & \text{otherwise} \end{cases}$$
$$\qquad (11)$$

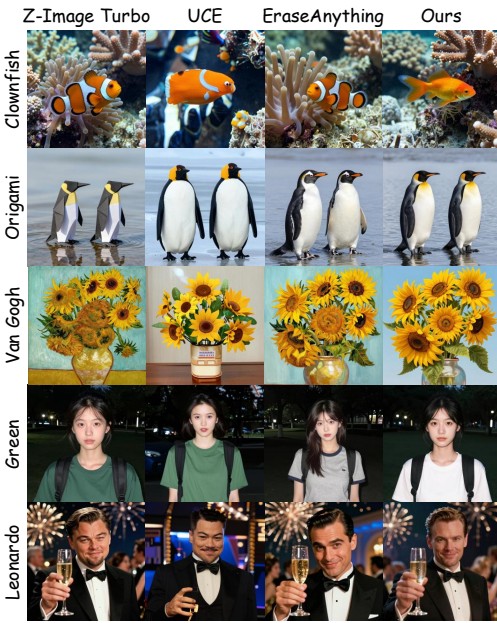

Figure 5. **Single-concept erasure.** Our method removes both concrete and abstract concepts while preserving image quality with minimal collateral changes. In contrast, UCE severely distorts images and introduces strong artifacts, whereas EraseAnything may fail to erase or causes noticeable semantic shifts.

where

$$\lambda_t^* = \frac{-\nabla\mathcal{L}_{er}(\theta_t) \cdot \nabla\mathcal{L}_{pr}(\theta_t) - \varepsilon}{\|\nabla\mathcal{L}_{pr}(\theta_t)\|^2}. \quad (12)$$

Due to space limit, we provide full derivation from Eq. 9 to Eq. 12 in Appendix B.1 and B.2. Here, $\lambda_t^*$ acts as a dynamic gatekeeper. If the erasure gradient attempts to hurt preservation capability beyond $\varepsilon$, $\lambda_t^*$ becomes positive, effectively projecting the update onto a safe subspace, as illustrated in Fig. 4.

However, we find that explicitly computing $\lambda_t^*$ requires two independent backward passes to obtain $\nabla\mathcal{L}_{er}$ and $\nabla\mathcal{L}_{pr}$, doubling the training cost. To make this theoretically grounded approach computationally feasible, we employ an implicit approximation key to our adaptive modulation. Specifically, instead of directly solving for the exact $\lambda_t$ at every step, we approximate the minimum of the dual problem Eq. 10 by iteratively updating $\lambda_t$ via gradient descent:

$$\lambda_{t+1} = \lambda_t - \beta\nabla_{\lambda_t}\mathcal{L}_t(\lambda_t), \quad (13)$$

where $\beta$ controls the sensitivity of the weight adjustment. However, calculating $\nabla_{\lambda_t}\mathcal{L}_t$ still needs the gradient $\nabla\mathcal{L}_{pr} \cdot \mathbf{d}_t$, which brings us back to the original computational bottleneck. To resolve this, we approximate this dot

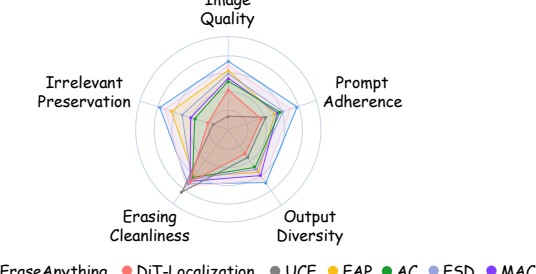

Figure 6. **User study.** An interface is built to compare generated images from various erasure methods on Z-Image Turbo (see Appendix E for details). Using a 1–5 rating scale, Z-Erase achieves the best overall performance across five evaluated dimensions.

product $\nabla\mathcal{L}_{pr} \cdot \mathbf{d}_t$ using the first-order Taylor expansion:

$$\begin{aligned}
\nabla_{\lambda_t}\mathcal{L}_t(\lambda_t) &= \nabla\mathcal{L}_{pr}(\theta_t) \cdot (\nabla\mathcal{L}_{er}(\theta_t) + \lambda_t\nabla\mathcal{L}_{pr}(\theta_t)) + \varepsilon \\
&= \nabla\mathcal{L}_{pr}(\theta_t) \cdot \mathbf{d}_t + \varepsilon \\
&\approx \frac{1}{\alpha}\left(\mathcal{L}_{pr}(\theta_t) - \mathcal{L}_{pr}(\theta_{t+1})\right) + \varepsilon.
\end{aligned} \quad (14)$$

Here, $\alpha$ is the learning rate of the model. This step is crucial: it translates the expensive geometric projection (dot product of gradients) into a cheap scalar observation (change in loss values). As a result, we obtain an approximate method for solving $\lambda_t$ without extra backpropagation:

$$\lambda_{t+1} = \lambda_t - \beta\tilde{g}_t, \text{ where } \tilde{g}_t = \frac{1}{\alpha}\left(\mathcal{L}_{pr}(\theta_{t-1}) - \mathcal{L}_{pr}(\theta_t)\right) + \varepsilon. \quad (15)$$

This creates a self-regulating loop: if the model violates the preservation constraint (*i.e.*, $\mathcal{L}_{pr}$ increases significantly such that $\tilde{g}_t < 0$), $\lambda$ increases, forcing the model to prioritize preservation in the next unified backward pass; otherwise, $\lambda$ decays, allowing aggressive concept erasure.

Notably, a major advantage of this algorithm is that it provides a theoretical bound on how much the model's original capabilities will shift. We can prove the degradation of the preservation capability is bounded by:

$$\mathcal{L}_{pr}(\theta_t) - \mathcal{L}_{pr}(\theta_0) \lesssim \mathcal{O}(t\,\varepsilon\,\alpha). \quad (16)$$

Detailed proofs are provided in Appendix B.3. Here, $\varepsilon$ *directly governs the trade-off*: smaller values enforce tighter preservation, while larger ones allow controlled deviations to erase stubborn concepts. Despite using an efficient first-order approximation, we rigorously prove in Appendix C that this strategy guarantees asymptotic convergence to a Pareto stationary point, ensuring a principled balance. The full training procedure is summarized in Algorithm 1.

*Table 1.* **Evaluation on NSFW erasure.** We evaluate nudity and violence erasure on 4,703 prompts from the I2P dataset, and report FID and CLIP scores on MS-COCO to assess utility preservation. Results of the original Z-Image Turbo are presented for reference.

| METHOD | DETECTED NUDITY | | | | MS-COCO 10K | | DETECTED VIOLENCE ↓ | MS-COCO 10K | |
| --- | --- | --- | --- | --- | --- | --- | --- | --- | --- |
| | Common | Female | Male | Total ↓ | FID ↓ | CLIP ↑ | | FID ↓ | CLIP ↑ |
| AC (Model-Based) (Kumari et al., 2023) | 245 | 35 | 53 | 333 | 29.30 | 29.05 | 402 | 32.47 | 29.89 |
| AC (Noise-Based) (Kumari et al., 2023) | 270 | 38 | 55 | 363 | 30.07 | 28.16 | 496 | 33.12 | 28.73 |
| ESD-x (Gandikota et al., 2023) | 279 | 42 | 60 | 381 | 31.44 | 27.65 | 777 | 33.79 | 27.48 |
| ESD-u (Gandikota et al., 2023) | 260 | 41 | 61 | 362 | 31.96 | 27.13 | 668 | 34.22 | 26.80 |
| EAP (Bui et al., 2024) | 190 | 29 | 49 | 268 | 27.32 | 30.71 | 447 | 28.30 | 30.15 |
| MACE (Lu et al., 2024) | 136 | 35 | 28 | 199 | 28.75 | 30.48 | 382 | 29.63 | 29.86 |
| UCE (Gandikota et al., 2024) | 129 | 15 | 13 | **157** | 38.38 | 21.66 | 561 | 36.02 | 27.02 |
| Meta-Unlearning (Gao et al., 2025b) | 346 | 53 | 56 | 455 | 26.91 | 30.83 | 845 | **27.36** | 31.08 |
| EraseAnything (Gao et al., 2025a) | 206 | 43 | 45 | 294 | 26.63 | 31.10 | - | - | - |
| Minimalist CE (Zhang et al., 2025) | 291 | 38 | 47 | 376 | 27.23 | 30.16 | 702 | 27.82 | 30.06 |
| DiT Localization (Zarei et al., 2025) | 264 | 40 | 52 | 356 | 32.65 | 27.12 | 649 | 32.65 | 27.12 |
| Ours | 109 | 29 | 23 | 161 | **26.46** | **31.25** | **324** | 27.62 | **31.17** |
| Z-Image Turbo | 486 | 100 | 63 | 649 | 26.33 | 31.47 | 1086 | 26.33 | 31.47 |

*Table 2.* **Evaluation on celebrity erasure.** CLIP-based classification accuracy are reported on the erased identities ($\text{ACC}_e$, efficacy) and unaffected identities ($\text{ACC}_{ir}$, specificity). The combined performance is summarized by $H_a = \text{ACC}_{ir} - \text{ACC}_e$, where higher values indicate a better balance between erasure and preservation. FID and CLIP are results based on MS-COCO 10K dataset.

| METHOD | $\text{ACC}_e$ ↓ | $\text{ACC}_{ir}$ ↑ | $H_a$ ↑ | FID ↓ | CLIP ↑ |
| --- | --- | --- | --- | --- | --- |
| AC (Model-Based) | 27.95 | 27.10 | -0.85 | 31.85 | 28.22 |
| AC (Noise-Based) | 27.34 | 26.88 | -0.46 | 32.46 | 27.93 |
| ESD-x | 27.66 | 27.20 | -0.46 | 33.15 | 27.41 |
| ESD-u | 25.13 | 26.68 | 1.55 | 34.06 | 26.95 |
| MACE | 24.06 | **28.52** | 4.46 | 30.79 | **30.17** |
| UCE | **21.91** | 25.70 | 3.79 | 52.28 | 23.70 |
| EraseAnything | 23.25 | 27.36 | 3.71 | 27.86 | 29.65 |
| DiT-Localization | 24.43 | 23.41 | -1.02 | 43.27 | 26.76 |
| Ours | 23.52 | 28.34 | **4.82** | 27.83 | 30.06 |
| Z-Image Turbo | 33.91 | 32.24 | - | 26.33 | 31.47 |

*Table 3.* **Evaluation on erasing specific category**: **Entity** (*e.g.,* church), **Artistic Style** (*e.g.,* Claude Monet) and **Abstraction** (*e.g.,* color). We report erasure efficacy ($\text{ACC}_e$) and balance score ($H_a$) here due to space limit. The full metrics are in Appendix D.2.

| METHOD | ENTITY | | ART STYLE | | ABSTRACTION | |
| --- | --- | --- | --- | --- | --- | --- |
| | $\text{ACC}_e$ ↓ | $H_a$ ↑ | $\text{ACC}_e$ ↓ | $H_a$ ↑ | $\text{ACC}_e$ ↓ | $H_a$ ↑ |
| AC (Model-Based) | 24.6 | 2.0 | 28.6 | -4.2 | 26.9 | -1.1 |
| AC (Noise-Based) | 25.5 | 0.9 | 29.3 | -5.5 | 27.3 | -1.1 |
| ESD-x | 23.3 | 5.3 | 28.5 | -3.8 | 26.5 | -0.8 |
| ESD-u | 21.8 | 6.0 | 27.8 | -3.3 | 25.7 | -0.4 |
| MACE | 23.2 | 4.3 | 26.6 | -0.8 | 25.1 | 3.4 |
| UCE | 20.1 | 0.5 | **21.7** | -1.8 | **19.7** | 0.6 |
| EraseAnything | 24.1 | 3.5 | 25.9 | 0.2 | 25.6 | 3.4 |
| DiT-Localization | **17.5** | **11.2** | 31.2 | -7.9 | 28.6 | -2.8 |
| Ours | 23.1 | 7.4 | 26.1 | **0.8** | 24.3 | **4.7** |
| Z-Image Turbo | 24.7 | - | 33.6 | - | 30.5 | - |

# 5. Experiments

## 5.1. Implementation Details

Currently, the only publicly available pure single-stream T2I diffusion models are Z-Image Turbo[1] and HunyuanImage-3.0[2]. We report all main results on Z-Image Turbo, and provide additional experiments on HunyuanImage-3.0 in Appendix G. Our experiments adopt the flow-matching Euler sampler with 9 steps and AdamW optimizer (Loshchilov et al., 2017) optimizer for 1,000 steps, with learning rate $\alpha = 0.001$, $\beta = 0.1$, and erasing guidance $\eta = 2$.

Notably, all fine-tuning baselines rely on our *Stream Disentangled Concept Erasure Framework* (Eq. 4) on single-stream models to function properly and prevent generation collapse. For more details, please refer to Appendix D.

---

[1]https://huggingface.co/Tongyi-MAI/Z-Image Turbo
[2]https://huggingface.co/tencent/HunyuanImage-3.0

## 5.2. Results

**NSFW Erasure.** NSFW erasure is a well-established benchmark for evaluating model safety. We assess our method on a comprehensive set of 4,703 prompts from the Inappropriate Image Prompt (I2P) dataset (Schramowski et al., 2023), focusing on **nudity** and **violence**. For nudity detection, we adopt NudeNet (Bedapudi, 2019) with a detection threshold of 0.6, and for violence, we employ the Q16-classifier (Schramowski et al., 2022). To further evaluate the specificity of our method on regular content, we randomly select 10,000 captions from MS-COCO dataset (Lin et al., 2014). Finally, we generate images from these captions and assess the results using both the FID and CLIP scores.

Table 1 presents our results compared with the current state-of-the-art algorithms. Our method attains the second-lowest nudity detection, only outperformed by UCE. Yet, our method maintains strong FID and CLIP scores, indicating minimal degradation on regular content, while UCE

*Table 4.* **Evaluation against adversarial attacks.** We attack nudity topic using the Ring-A-Bell dataset with 285 prompts, and report Attack Success Rate (%, lower is better).

| METHOD | W/O ATTACK | R-A-B | REFLUX | UNLEARNDIFFATK Step = 0 | UNLEARNDIFFATK Steps = 0,1,2 |
|---|---|---|---|---|---|
| DiT-Localization | 57.89 | 58.94 | 74.73 | 59.64 | 60.35 |
| EraseAnything | 14.38 | 26.67 | 43.85 | **23.50** | **25.61** |
| Token-Zeroing | 21.75 | 82.81 | 92.85 | 72.50 | 76.49 |
| Ours | **10.52** | **24.91** | **42.81** | 24.56 | 25.61 |
| Z-Image Turbo | 83.15 | 92.63 | 96.84 | 82.45 | 85.96 |

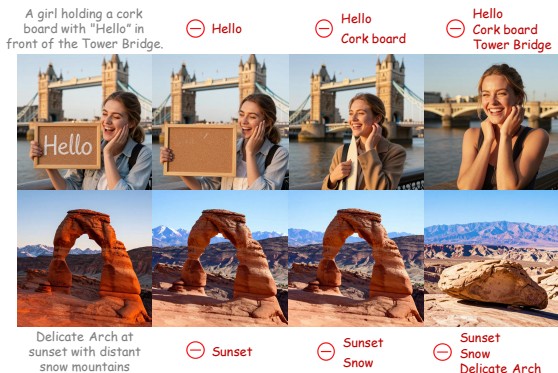

*Figure 7.* **Multi-concept erasure.** Multiple concepts can be erased simultaneously via weighted LoRA averaging (see Appendix F).

shows a sharp decline in utility according to these metrics. Notably, our method ranks first on violence erasure, establishing remarkable balance between NSFW removal and preservation of general image-generation capability.

**Celebrity Erasure.** Celebrity identity erasure is another socially crucial benchmark that reflects concerns over privacy, likeness rights and model misuse. We select a subset from CelebA (Liu et al., 2018), removing identities that Z-Image Turbo fails to reproduce faithfully. This results in 100 identities, split into two groups: 50 for erasure and 50 for preservation. We adopt the evaluation metrics in Table 2. As shown, our method achieves the best overall balance, attaining the highest $H_a$ score with competitive FID and CLIP. These results highlight the potential of our approach to support privacy-aware content regulation without compromising generative utility.

**Miscellaneousness Erasure.** We evaluate our method on 3 broader categories: **Entity**, **Artistic Style**, and **Abstraction**. Here, we choose 10 concept for each category (Please check Appendix D.2 for the full list of concepts) and adopt the metrics described in Table 3. The findings show that our approach remains consistently effective across diverse categories, with particularly stronger performance on abstract and global concepts. As shown in Fig. 5 and 7, our method reliably removes a wide range of concepts (including multiple concepts simultaneously) while introducing only minimal visual disturbance, whereas prior approaches often fail to erase or produce artifacts that undermine utility.

*Table 5.* **Ablation study on erasing nudity.** We ablate three loss terms used in our method on 4,703 prompts from the I2P dataset.

| CONFIG | DETECTED NUDITY ↓ | FID ↓ | CLIP ↑ |
|---|---|---|---|
| $\mathcal{L}_{erase}$ | 206 | 29.65 | 29.86 |
| $\mathcal{L}_{erase} + \mathcal{L}_{attn}$ | **148** | 30.04 | 29.12 |
| $\mathcal{L}_{erase} + \mathcal{L}_{pr}$ | 235 | **26.42** | **31.28** |
| $\mathcal{L}_{attn} + \mathcal{L}_{pr}$ | 314 | 27.15 | 31.16 |
| FULL | 161 | 26.46 | 31.25 |
| Z-Image Turbo | 649 | 26.33 | 31.47 |

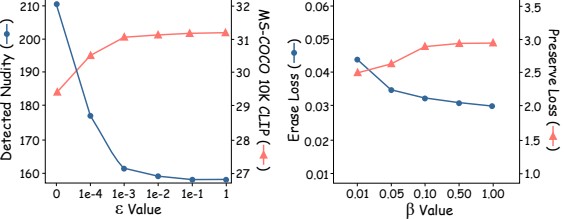

*Figure 8.* **Ablation study on hyperparameters $\varepsilon$ and $\beta$.** With our *Lagrangian-Guided Adaptive Erasure Modulation* algorithm, erasure and preservation become monotonically controllable.

**User Study.** We further assess human perceptual erasure quality through a five-dimensional user study. The first two dimensions (Erasing Cleanliness and Irrelevant Preservation) are evaluated using concepts from Entity, Artistic Style, and Abstraction, with images generated under same seeds for fair comparison. Our study involves 30 non-artist participants, each providing an average of 50 responses. As shown in Fig. 6, our method demonstrates consistently strong performance across all dimensions, positioning Z-Erase as a well-rounded solution for perceptual concept erasure. Additional study details are provided in Appendix E.

**Robustness under Adversarial Prompt Attacks.** To further evaluate robustness against adversarial attacks, we apply Ring-A-Bell (Tsai et al., 2024), UnlearnDiffAtk (Zhang et al., 2024), and ReFlux (Jiang et al., 2025) to attack the nudity concept. As shown in Table 4, the token-zeroing trick discussed in Section 3 is highly vulnerable to prompt perturbation, whereas our approach remains substantially more robust across most attack settings.

### 5.3. Ablation Study

**Ablation on Stream Disentangled Concept Erasure Framework.** We first validate the structural necessity of our framework. As visualized in Appendix Fig. 15, naive fine-tuning on single-stream models consistently causes generation collapse regardless of layer selection, proving our framework is a prerequisite for stable erasure. Within this safe subspace, we further ablate our loss terms in Table 5. $\mathcal{L}_{erase}$ effectively removes concepts but degrades quality, while $\mathcal{L}_{attn}$ enhances erasure precision. Crucially, $\mathcal{L}_{pr}$ serves as a stabilizer for visual utility. Combining all three

yields the optimal equilibrium between safety and quality.

**Ablation on Lagrangian-Guided Adaptive Erasure Modulation.** We then examine the optimization dynamics by varying tolerance $\varepsilon$ and learning rate $\beta$ in Fig. 8. Consistent with Section 4.2, a smaller $\varepsilon$ favors preservation at the cost of weaker erasure. We thus select $\varepsilon = 0.001$ for balance. In contrast, our method shows strong robustness to $\beta$ with negligible performance fluctuations, verifying that convergence is stable to step size. We set $\beta = 0.1$ as a practical choice.

## 6. Conclusion

In this work, we present **Z-Erase**, the first concept erasure method for single-stream models. To address the inherent architectural entanglement, we present *Stream Disentangled Concept Erasure Framework*, which structurally decouples parameter updates to prevent collapse, and *Lagrangian-Guided Adaptive Erasure Modulation* algorithm, which rigorously navigates the erasure-preservation trade-off within a highly sensitive latent. Experiments across diverse tasks demonstrate the effectiveness and versatility of our method.

## Acknoledgement

This work was funded by Frontier Technologies R&D Program of Jiangsu under Grant No. BF2025012 and by Beijing Advanced Innovation Center for Future Blockchain and Privacy Computing.

## Impact Statement

This work aims to advance the safety and reliability of modern text-to-image generative models by enabling effective concept erasure in emerging single-stream diffusion transformers. As these unified architectures are increasingly adopted due to their efficiency and performance, understanding how to responsibly control and remove undesired concepts becomes an important technical challenge. Our method provides a principled framework for mitigating the generation of harmful, sensitive, or unwanted content while preserving the general utility of the model.

We expect this work to have a positive societal impact by supporting safer deployment of generative models in real-world applications, such as content moderation, privacy protection, and compliance with ethical and legal requirements. More broadly, it contributes technical insights toward building controllable and responsible generative systems.

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

## A. Single-Stream Transformer Architecture

In our research, we adopt the emerging single-stream diffusion transformer as our foundational architecture. This paradigm, exemplified by recent state-of-the-art open-source models Z-Image Turbo (Team, 2025) and HunyuanImage-3.0 (Cao et al., 2025), represents a radical departure from the architectural conventions of both SD v1.x (Rombach et al., 2022) and the recent Flux series (Labs et al., 2025).

As highlighted in Section 3, the evolution of T2I architectures has progressively moved towards higher modality fusion. SD v1.x models utilize a *U-Net backbone* where visual features are processed in a dedicated path, and textual information is injected via distinct cross-attention layers. This separation allows for easy localization of concept-specific parameters. Flux introduces a *dual-stream backbone* where text and image tokens are processed by separate weights ($\mathbf{W}_{txt}$ and $\mathbf{W}_{img}$) for the first half of the network before being concatenated. While more unified than SD v1.x, it still maintains explicit separation in the early stages.

In contrast, the pure single-stream architecture (as shown in Fig. 9) abolishes these boundaries entirely. As depicted in the diagram, text embeddings (from large language models or vision language models) and image patches (from VAE latents) are mapped into a common embedding space via linear projections. Crucially, they are concatenated into a *Unified Input Sequence* before entering the transformer blocks. Within these blocks, the self-attention mechanism operates on this mixed sequence using *shared* projection weights ($\mathbf{W}_Q$, $\mathbf{W}_K$, $\mathbf{W}_V$) for both modalities.

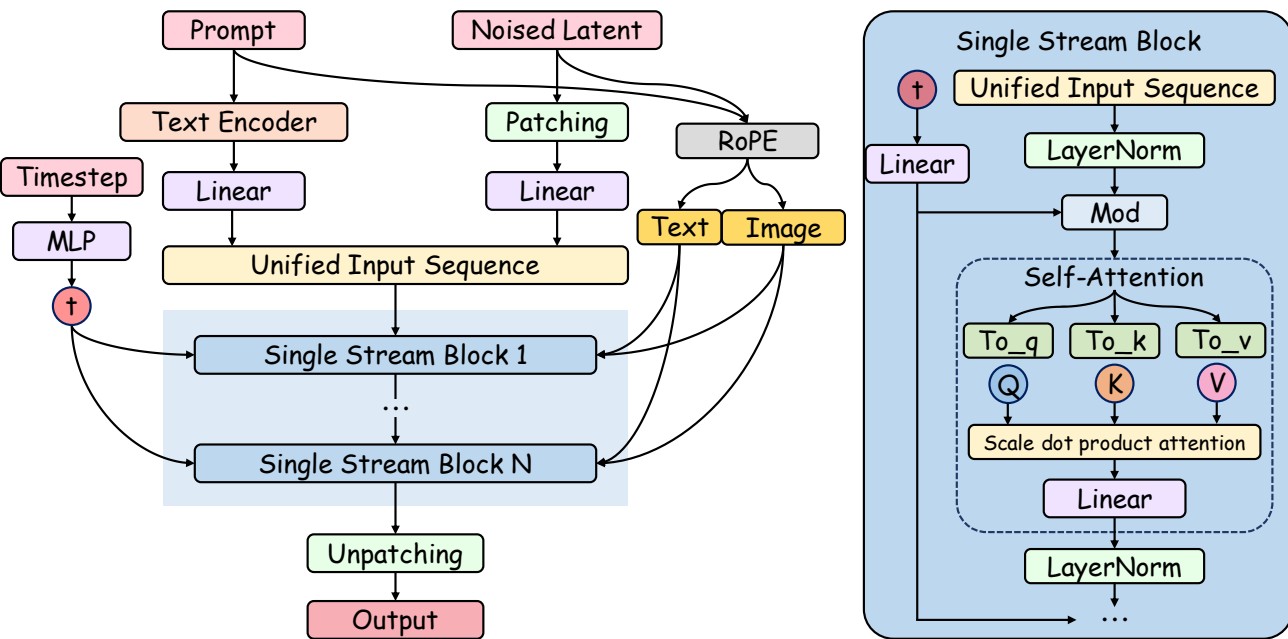

*Figure 9.* **Abstract illustration of the Single-Stream DiT architecture.** The workflow begins with two parallel pathways: the prompt is processed by a text encoder, while the image latent is patched and projected similarly. These tokens are concatenated into a *Unified Input Sequence*, which serves as the input to a stack of $N$ *Single Stream Blocks*. Global conditioning information (*e.g.*, Timestep) is injected into each block, influencing the layer normalization and attention scaling. Inside a Single Stream Block, the core *Self-Attention* mechanism processes the sequence using shared Query ($Q$), Key ($K$), and Value ($V$) projections for both text and image tokens.

This "Grand Unification" means that differentiation between text and image processing is no longer structural but semantic, driven solely by the token embeddings themselves. While this design maximizes parameter efficiency and generative quality, it introduces the *entanglement* challenge we solve in Z-Erase: modifying weights to erase a text concept inherently risks corrupting the shared image generation mechanics.

## B. Derivation Details of Lagrangian-Guided Adaptive Erasure Modulation

In this section, we present the complete derivations and proofs that are omitted from the main text due to space limitations.

### B.1. Derivation of the dual problem

To begin with, recall that the primal update in Eq. 9 jointly optimizes a high-dimensional descent direction $\mathbf{d}_t$ while enforcing a preservation constraint through an inner product term:

$$\max_{\mathbf{d}_t} \nabla\mathcal{L}_{er}(\theta_t) \cdot \mathbf{d}_t - \frac{1}{2}\|\mathbf{d}_t\|^2, \ \text{s.t.} \nabla\mathcal{L}_{pr}(\theta_t) \cdot \mathbf{d}_t \geq -\varepsilon, \tag{17}$$

Directly solving the primal form is undesirable for two reasons. First, the constraint only depends on the projection of $\mathbf{d}_t$ onto $\nabla\mathcal{L}_{pr}$, yet the optimization is carried out in the full parameter space, which obscures the proximal trade-off between erasure and preservation and offers no explicit control over the constraint (*e.g.*, $\varepsilon$). Second, similar constrained gradient formulations in robust optimization and adversarial training (Sener & Koltun, 2019; Yu et al., 2020; Zhou et al., 2025) have shown that converting the primal into its dual leads to substantially more stable updates and interpretable balancing coefficients. Motivated by these observations, we follow the standard primal-dual approach and rewrite Eq. 9 into its dual form via a Lagrangian relaxation. This reparameterizes the update using a scalar dual variable $\lambda_t \geq 0$ and naturally leads to the closed-form solution for $\mathbf{d}_t$ once the dual objective is minimized. We provide the complete derivation below.

*Proof.* We start by constructing the Lagrangian function associated with the primal problem. The Lagrangian can be written as:

$$\mathcal{L}(\mathbf{d}_t, \lambda_t) = \nabla\mathcal{L}_{er}(\theta_t) \cdot \mathbf{d}_t - \frac{1}{2}\|\mathbf{d}_t\|^2 + \lambda_t \left(\varepsilon + \nabla\mathcal{L}_{pr}(\theta_t) \cdot \mathbf{d}_t\right), \tag{18}$$

where $\lambda_t \geq 0$ is the Lagrange multiplier corresponding to the constraint in the primal problem.

To find the dual objective, we first need to minimize the Lagrangian *w.r.t* $\mathbf{d}_t$. Taking the gradient of $\mathcal{L}(\mathbf{d}_t, \lambda_t)$ *w.r.t* $\mathbf{d}_t$ and setting it to zero, we have:

$$\nabla_{\mathbf{d}_t}\mathcal{L}(\mathbf{d}_t, \lambda_t) = \nabla\mathcal{L}_{er}(\theta_t) + \lambda_t\nabla\mathcal{L}_{pr}(\theta_t) - \mathbf{d}_t = 0. \tag{19}$$

Solving for $\mathbf{d}_t$, we obtain:

$$\mathbf{d}_t = \nabla\mathcal{L}_{er}(\theta_t) + \lambda_t\nabla\mathcal{L}_{pr}(\theta_t). \tag{20}$$

Substituting this back into the Lagrangian, we then get the dual function:

$$\mathcal{L}_t(\lambda_t) = \mathcal{L}(\mathbf{d}_t, \lambda_t) = \nabla\mathcal{L}_{er}(\theta_t) \cdot (\nabla\mathcal{L}_{er}(\theta_t) + \lambda_t\nabla\mathcal{L}_{pr}(\theta_t)) - \frac{1}{2}\|\nabla\mathcal{L}_{er}(\theta_t) + \lambda_t\nabla\mathcal{L}_{pr}(\theta_t)\|^2 + \lambda_t\varepsilon. \tag{21}$$

By expanding and simplifying, the dual objective finally becomes:

$$\mathcal{L}_t(\lambda_t) = \frac{1}{2}\|\nabla\mathcal{L}_{er}(\theta_t) + \lambda_t\nabla\mathcal{L}_{pr}(\theta_t)\|^2 + \lambda_t\varepsilon, \tag{22}$$

which is the dual objective in Eq. 10. $\square$

### B.2. Derivation of the closed-form solution for the optimal direction $\mathbf{d}_t^*$

*Proof.* To find the closed-form solution for the optimal direction $\mathbf{d}_t^*$, we first minimize the dual objective $\mathcal{L}_t(\lambda_t)$ *w.r.t* $\lambda_t$. Taking the derivative of $\mathcal{L}_t(\lambda_t)$ *w.r.t* $\lambda_t$ and setting it to zero, we then get:

$$\frac{\partial\mathcal{L}_t(\lambda_t)}{\partial\lambda_t} = \nabla\mathcal{L}_{pr}(\theta_t) \cdot (\nabla\mathcal{L}_{er}(\theta_t) + \lambda_t\nabla\mathcal{L}_{pr}(\theta_t)) + \varepsilon = 0. \tag{23}$$

Solving for $\lambda_t$, we obtain:

$$\lambda_t^* = \frac{-\nabla\mathcal{L}_{pr}(\theta_t) \cdot \nabla\mathcal{L}_{er}(\theta_t) - \varepsilon}{\|\nabla\mathcal{L}_{pr}(\theta_t)\|^2}. \tag{24}$$

Substituting $\lambda_t^*$ back into the expression for $\mathbf{d}_t^*$, we find the optimal update direction $\mathbf{d}_t^*$ as:

$$\mathbf{d}_t^* = \begin{cases} \nabla\mathcal{L}_{er}(\theta_t) + \lambda_t^*\nabla\mathcal{L}_{pr}(\theta_t), & \text{if } \lambda_t^* > 0, \\ \nabla\mathcal{L}_{er}(\theta_t), & \text{if } \lambda_t^* \leq 0. \end{cases} \tag{25}$$

This is the closed-form solution for $\mathbf{d}_t^*$ stated in Eq. 11. $\square$

### B.3. Derivation of upper bound on preservation degradation

*Proof.* We first assume that the preservation loss $\mathcal{L}_{pr}$ is $\mathbf{G}$-smooth, meaning that its gradient is $\mathbf{G}$-Lipschitz:

$$\|\nabla\mathcal{L}_{pr}(\theta) - \nabla\mathcal{L}_{pr}(\theta')\| \leq \mathbf{G}\|\theta - \theta'\|, \tag{26}$$

for some constant $\mathbf{G} > 0$. By the standard smoothness inequality, we obtain:

$$\mathcal{L}_{pr}(\theta_{t+1}) \leq \mathcal{L}_{pr}(\theta_t) + \nabla\mathcal{L}_{pr}(\theta_t)(\theta_{t+1} - \theta_t) + \frac{\mathbf{G}}{2}\|\theta_{t+1} - \theta_t\|^2. \tag{27}$$

Substituting $\theta_{t+1} = \theta_t - \alpha\mathbf{d}_t$ (where $\alpha$ is the learning rate of $\theta$) gives:

$$\mathcal{L}_{pr}(\theta_{t+1}) - \mathcal{L}_{pr}(\theta_t) \leq -\alpha\nabla\mathcal{L}_{pr}(\theta_t)\mathbf{d}_t + \frac{\mathbf{G}\alpha^2}{2}\|\mathbf{d}_t\|^2. \tag{28}$$

Using the primal constraint $\nabla\mathcal{L}_{pr}(\theta_t) \cdot \mathbf{d}_t \geq -\varepsilon$, we obtain:

$$\mathcal{L}_{pr}(\theta_{t+1}) - \mathcal{L}_{pr}(\theta_t) \leq \alpha\varepsilon + \frac{\mathbf{G}\alpha^2}{2}\|\mathbf{d}_t\|^2 \lesssim \alpha\varepsilon, \tag{29}$$

where the final approximation holds when $\alpha$ is sufficiently small such that the quadratic term becomes negligible. Summing over $0$ to $t$, we get:

$$\mathcal{L}_{pr}(\theta_t) - \mathcal{L}_{pr}(\theta_0) \lesssim \mathcal{O}(t\,\varepsilon\,\alpha), \tag{30}$$

which is the upper bound in Eq. 16. $\qquad\square$

## C. Convergence Analysis of Lagrangian-Guided Adaptive Erasure Modulation

In this section, we provide a theoretical analysis of the convergence properties of the proposed **Lagrangian-Guided Adaptive Erasure Modulation** algorithm. Our primary goal is to demonstrate that Algorithm 1, despite employing a specific first-order Taylor approximation for computational efficiency (Eq. 14), successfully converges to a **Pareto stationary point**. This implies that the algorithm finds a solution where the erasure objective $\mathcal{L}_{er}$ cannot be further optimized without violating the preservation constraint defined by $\mathcal{L}_{pr}$ and tolerance $\varepsilon$.

Formally, the optimization problem in Z-Erase can be modeled as a constrained optimization problem: $\min_\theta \mathcal{L}_{er}(\theta)$ s.t. the preservation degradation is bounded by $\varepsilon$. Our algorithm solves this via a dynamic Lagrangian method. The core challenge in the analysis lies in the coupled dynamics of the model parameters $\theta_t$ and the constraint weight $\lambda_t$, particularly given that $\lambda_t$ is updated via an **implicit approximation** (without calculating the full Hessian or second-order information).

To rigorously prove the convergence, our analysis proceeds in three main steps:

1. **Approximation Bound**: We first establish that the implicitly estimated gradient for updating $\lambda_t$ (derived from the loss difference in Eq. 14) serves as a valid proxy for the true gradient, with the error bounded by the learning rate $\alpha$.

2. **Dynamic Regret Analysis**: We treat the update of the dynamic weight $\lambda_t$ as an online learning process. By leveraging results from Online Gradient Descent (OGD), we prove that the accumulation of errors (regret) due to the dynamic nature of $\lambda_t$ and $\beta$ is sub-linear and controllable. This answers why the dynamic variation of $\beta$ (or $\lambda$) does not prevent the system from settling.

3. **Convergence to Stationarity**: Finally, combining the smoothness of the objective functions and the bounded regret, we prove that the gradient of the total objective tends to zero, ensuring the method converges to a valid stationary solution.

### C.1. Approximation bound

Our analysis relies on the following standard assumptions regarding the erasure objective $\mathcal{L}_{er}$ and the preservation objective $\mathcal{L}_{pr}$.

**Assumption C.1** (Boundedness and Lipschitz Continuity)**.** For both objective functions $f \in \{\mathcal{L}_{er}, \mathcal{L}_{pr}\}$, we assume they are bounded below and $\mathbf{L}$-Lipschitz continuous. That is, for any parameters $\theta_1, \theta_2$, there exists a constant $\mathbf{L} > 0$ such that $\|\nabla f(\theta_1)\| \leq \mathbf{L}$ and $|f(\theta_1) - f(\theta_2)| \leq \mathbf{L}\|\theta_1 - \theta_2\|$.

**Assumption C.2** (**G**-Smoothness)**.** The objective functions are differentiable and their gradients are **G**-Lipschitz continuous (**G**-Smooth). Specifically, there exists a constant **G** $> 0$ such that:

$$\|\nabla f(\theta_1) - \nabla f(\theta_2)\| \leq \mathbf{G}\|\theta_1 - \theta_2\|. \tag{31}$$

A key consequence of **G**-smoothness is the descent lemma (or quadratic upper bound):

$$|f(\theta_2) - f(\theta_1) - \nabla f(\theta_1)^\top(\theta_2 - \theta_1)| \leq \frac{\mathbf{G}}{2}\|\theta_2 - \theta_1\|^2. \tag{32}$$

Recall that the core of our **Lagrangian-Guided Adaptive Erasure Modulation** involves solving the dual problem (Eq. 10) with respect to $\lambda_t$. The update rule for $\lambda_t$ depends on the gradient of the dual objective $\mathcal{L}_t(\lambda_t)$. From Eq. 23 (in the preceding derivation), the *true gradient* of the dual objective *w.r.t* $\lambda_t$ is given by:

$$g_t \triangleq \nabla_{\lambda_t}\mathcal{L}_t(\lambda_t) = \nabla\mathcal{L}_{pr}(\theta_t) \cdot \mathbf{d}_t + \varepsilon, \tag{33}$$

where $\mathbf{d}_t$ is the update direction for model parameters. However, computing $\nabla\mathcal{L}_{pr}(\theta_t) \cdot \mathbf{d}_t$ exactly requires two backward passes. To improve efficiency, Z-Erase utilizes an **approximate gradient** update (Eq. 14) using the loss difference from the first-order Taylor approximation:

$$\tilde{g}_t \triangleq \frac{1}{\alpha}(\mathcal{L}_{pr}(\theta_{t-1}) - \mathcal{L}_{pr}(\theta_t)) + \varepsilon, \tag{34}$$

where $\theta_t = \theta_{t-1} - \alpha\mathbf{d}_{t-1}$.

The following lemma establishes that this implicit approximation is a theoretically valid proxy, as the approximation error is bounded by the model learning rate $\alpha$.

**Lemma C.3** (Approximation Error Bound)**.** *Under Assumptions C.1 and C.2, the difference between the true gradient $g_t$ and the approximate gradient $\tilde{g}_t$ used in Algorithm 1 is bounded by the step size $\alpha$:*

$$\|\tilde{g}_t - g_t\| \leq \frac{\mathbf{G}\alpha}{2}\|\mathbf{d}_{t-1}\|^2 \leq \mathcal{O}(\alpha). \tag{35}$$

*Proof.* Substituting the parameter update rule $\theta_t = \theta_{t-1} - \alpha\mathbf{d}_{t-1}$ into the smoothness inequality (Assumption C.2) for the preservation loss $\mathcal{L}_{pr}$, we have:

$$\mathcal{L}_{pr}(\theta_t) \leq \mathcal{L}_{pr}(\theta_{t-1}) + \nabla\mathcal{L}_{pr}(\theta_{t-1})^\top(-\alpha\mathbf{d}_{t-1}) + \frac{\mathbf{G}}{2}\| - \alpha\mathbf{d}_{t-1}\|^2. \tag{36}$$

Rearranging the terms to isolate the dot product $\nabla\mathcal{L}_{pr}(\theta_{t-1})^\top\mathbf{d}_{t-1}$:

$$\alpha\nabla\mathcal{L}_{pr}(\theta_{t-1})^\top\mathbf{d}_{t-1} \leq \mathcal{L}_{pr}(\theta_{t-1}) - \mathcal{L}_{pr}(\theta_t) + \frac{\mathbf{G}\alpha^2}{2}\|\mathbf{d}_{t-1}\|^2. \tag{37}$$

Similarly, using the lower bound property of smoothness, we can obtain:

$$\mathcal{L}_{pr}(\theta_t) \geq \mathcal{L}_{pr}(\theta_{t-1}) - \alpha\nabla\mathcal{L}_{pr}(\theta_{t-1})^\top\mathbf{d}_{t-1} - \frac{\mathbf{G}\alpha^2}{2}\|\mathbf{d}_{t-1}\|^2, \tag{38}$$

which implies:

$$\alpha\nabla\mathcal{L}_{pr}(\theta_{t-1})^\top\mathbf{d}_{t-1} \geq \mathcal{L}_{pr}(\theta_{t-1}) - \mathcal{L}_{pr}(\theta_t) - \frac{\mathbf{G}\alpha^2}{2}\|\mathbf{d}_{t-1}\|^2. \tag{39}$$

Combining these two inequalities, the absolute difference satisfies:

$$\left|(\mathcal{L}_{pr}(\theta_{t-1}) - \mathcal{L}_{pr}(\theta_t)) - \alpha\nabla\mathcal{L}_{pr}(\theta_{t-1})^\top\mathbf{d}_{t-1}\right| \leq \frac{\mathbf{G}\alpha^2}{2}\|\mathbf{d}_{t-1}\|^2. \tag{40}$$

Now, we explicitly compare the gradients. The error is:

$$\|\tilde{g}_t - g_t\| = \left| \left( \frac{1}{\alpha}(\mathcal{L}_{pr}(\theta_{t-1}) - \mathcal{L}_{pr}(\theta_t)) + \varepsilon \right) - (\nabla\mathcal{L}_{pr}(\theta_{t-1}) \cdot \mathbf{d}_{t-1} + \varepsilon) \right| \tag{41}$$

$$= \frac{1}{\alpha} \left| (\mathcal{L}_{pr}(\theta_{t-1}) - \mathcal{L}_{pr}(\theta_t)) - \alpha\nabla\mathcal{L}_{pr}(\theta_{t-1}) \cdot \mathbf{d}_{t-1} \right| \tag{42}$$

$$\leq \frac{1}{\alpha} \cdot \frac{\mathbf{G}\alpha^2}{2} \|\mathbf{d}_{t-1}\|^2 \tag{43}$$

$$= \frac{\mathbf{G}\alpha}{2} \|\mathbf{d}_{t-1}\|^2. \tag{44}$$

Since $\mathcal{L}_{er}$ and $\mathcal{L}_{pr}$ are Lipschitz continuous (Assumption C.1), the gradient $\nabla\mathcal{L}_{er}$ and $\nabla\mathcal{L}_{pr}$ are bounded, which implies the update direction $\mathbf{d}_t$ (a linear combination of gradients) is also bounded, *i.e.*, $\|\mathbf{d}_{t-1}\|^2 \leq C$. Thus, strictly speaking, $\|\tilde{g}_t - g_t\| \leq \mathcal{O}(\alpha)$. This concludes the proof that the implicit gradient approximation is asymptotically accurate as $\alpha \to 0$. □

### C.2. Dynamic regret analysis

Since the model parameters $\theta_t$ are updated at each step, the dual objective function $\mathcal{L}_t(\lambda)$ is actually time-varying. Consequently, the update of the constraint weight $\lambda_t$ via gradient ascent can be formulated as an Online Convex Optimization (OCO) problem. To prove convergence, we must bound the *dynamic regret*, which measures the accumulated loss difference between our algorithm's choice of $\lambda_t$ and the sequence of optimal comparators $\lambda_t^*$.

We define the dual objective at step $t$ following Eq. 10 as $\mathcal{L}_t(\lambda_t) = \frac{1}{2}\|\nabla\mathcal{L}_{er}(\theta_t) + \lambda_t\nabla\mathcal{L}_{pr}(\theta_t)\|^2 + \lambda_t\varepsilon$. Note that we aim to minimize this dual objective *w.r.t* $\lambda_t$. The dynamic regret is defined as:

$$\mathcal{R}_T \triangleq \sum_{t=1}^{T} \left( \mathcal{L}_t(\lambda_t) - \mathcal{L}_t(\lambda_t^*) \right), \tag{45}$$

where $\lambda_t^* = \arg\min_{\lambda_t \geq 0} \mathcal{L}_t(\lambda_t)$ is the optimal dual variable at step $t$ (derived in Eq. 12).

To bound this regret, we first establish a bound on the *total functional variation of the dual objective*, which quantifies how much the optimization landscape changes between steps.

**Lemma C.4** (Total Functional Variation). *Under Assumptions C.1 and C.2, let $D$ be the upper bound of the dual variable $\lambda$. The variation of the dual objective function between consecutive steps is bounded by the learning rate $\alpha$:*

$$\sum_{t=0}^{T-1} \sup_{\lambda\in[0,D]} |\mathcal{L}_{t+1}(\lambda) - \mathcal{L}_t(\lambda)| \leq C_1 T\alpha, \tag{46}$$

*where $C_1$ is a constant depending on Lipschitz constant $\mathbf{L}$, Smoothness constant $\mathbf{G}$, and $D$.*

*Proof.* For any $\lambda$, the variation is:

$$|\mathcal{L}_{t+1}(\lambda) - \mathcal{L}_t(\lambda)| = \left| \frac{1}{2}\|\nabla\mathcal{L}_{er}(\theta_{t+1}) + \lambda\nabla\mathcal{L}_{pr}(\theta_{t+1})\|^2 - \frac{1}{2}\|\nabla\mathcal{L}_{er}(\theta_t) + \lambda\nabla\mathcal{L}_{pr}(\theta_t)\|^2 \right| \tag{47}$$

$$= \frac{1}{2} \left| (\mathbf{v}_{t+1} + \mathbf{v}_t)^\top (\mathbf{v}_{t+1} - \mathbf{v}_t) \right|, \tag{48}$$

where we denote the composite gradient vector $\mathbf{v}_t(\lambda) \triangleq \nabla\mathcal{L}_{er}(\theta_t) + \lambda\nabla\mathcal{L}_{pr}(\theta_t)$. By L-Lipschitz continuity, $\|\mathbf{v}_t\| \leq \mathcal{L}(1+\lambda)$. By $\mathbf{G}$-smoothness, $\|\mathbf{v}_{t+1} - \mathbf{v}_t\| \leq \mathbf{G}(1+\lambda)\|\theta_{t+1} - \theta_t\| = \mathbf{G}(1+\lambda)\alpha\|\mathbf{d}_t\|$. Substituting these bounds:

$$|\mathcal{L}_{t+1}(\lambda) - \mathcal{L}_t(\lambda)| \leq \frac{1}{2} \cdot 2\mathbf{L}(1+\lambda) \cdot \mathbf{G}(1+\lambda)\alpha\|\mathbf{d}_t\| \tag{49}$$

$$\leq \alpha \cdot \mathbf{GL}(1+D)^2\|\mathbf{d}_t\|. \tag{50}$$

Since $\|\mathbf{d}_t\|$ is bounded (as gradients are bounded), let $C_1 = \mathbf{GL}(1+D)^2 \max \|\mathbf{d}_t\|$. Summing over $t$ yields the lemma. □

With the variation bound established, we invoke the standard dynamic regret bound for Online Gradient Descent (OGD) (Zinkevich, 2003; Jadbabaie et al., 2015).

**Theorem C.5** (Dynamic Regret Bound). *Suppose the updates for $\lambda_t$ follow the gradient descent rule with a fixed step size $\beta$ (as in Eq. 13), and model updates use a fixed learning rate $\alpha$. Using the approximate gradient $\tilde{g}_t$ where $\|\tilde{g}_t - g_t\| \leq \mathcal{O}(\alpha)$, the average dynamic regret is bounded by terms dependent on the step sizes and the horizon $T$:*

$$\frac{1}{T} \sum_{t=1}^{T} (\mathcal{L}_t(\lambda_t) - \mathcal{L}_t(\lambda_t^*)) \leq \mathcal{O}\left(\frac{1}{\beta T}\right) + \mathcal{O}(\beta) + \mathcal{O}(\alpha). \tag{51}$$

*As $T \to \infty$, the average regret converges to a neighborhood $\mathcal{O}(\alpha + \beta)$ determined by the chosen hyperparameters.*

*Proof.* (Sketch) The regret in online learning with inexact gradients can be decomposed into two components: one arising from the non-stationarity of the environment (tracking error) and one from the gradient approximation error. Standard results in Online Convex Optimization (OCO) with *constant step sizes* indicate that the tracking error is upper-bounded by $\mathcal{O}(\beta)$ (representing the system's reaction speed to dynamic changes) plus an initialization term decaying as $\mathcal{O}(\frac{1}{\beta T})$. Additionally, Lemma C.3 ensures that the gradient approximation error is bounded by $\mathcal{O}(\alpha)$ at each step. Averaging over $T$ steps, the total regret bound becomes $\mathcal{O}(\frac{1}{\beta T} + \beta + \alpha)$. This result provides sufficient theoretical justification for our practical implementation: while exact convergence to zero requires diminishing steps, establishing small fixed $\alpha$ and $\beta$ guarantees the system stabilizes within a tight, controlled neighborhood of the optimal Pareto front. $\square$

### C.3. Convergence to Pareto stationarity

With the dynamic regret bound established, we now prove the convergence of the model parameters $\theta$ to a stationary point. We construct a total composite objective function at each step $t$ defined by the dynamically tracked weight $\lambda_t$:

$$\mathcal{J}_t(\theta) \triangleq \mathcal{L}_{er}(\theta) + \lambda_t \mathcal{L}_{pr}(\theta). \tag{52}$$

Our goal is to show that the gradient of this composite objective vanishes, implying that the algorithm reaches a consistent solution.

**Theorem C.6** (Pareto Stationarity). *Under Assumptions C.1 and C.2, and given the diminishing step sizes $\alpha, \beta$ satisfying the conditions in Theorem C.5, the sequence of iterates generated by Algorithm 1 satisfies:*

$$\lim_{T \to \infty} \min_{t=1,\ldots,T} \|\nabla \mathcal{L}_{er}(\theta_t) + \lambda_t \nabla \mathcal{L}_{pr}(\theta_t)\|^2 = 0. \tag{53}$$

*Proof.* Consider the update rule $\theta_{t+1} = \theta_t - \alpha_t \mathbf{d}_t$. By the construction of our algorithm (Eq. 11), the update direction is the gradient of the Lagrangian:

$$\mathbf{d}_t = \nabla \mathcal{L}_{er}(\theta_t) + \lambda_t \nabla \mathcal{L}_{pr}(\theta_t). \tag{54}$$

Applying the $\mathbf{G}$-smoothness assumption to the composite function $\mathcal{J}_t(\theta)$ (noting that $\mathcal{J}_t$ is smooth since it is a linear combination of smooth functions):

$$\mathcal{J}_t(\theta_{t+1}) \leq \mathcal{J}_t(\theta_t) + \nabla \mathcal{J}_t(\theta_t)^\top (\theta_{t+1} - \theta_t) + \frac{\mathbf{G}(1 + \lambda_t)}{2} \|\theta_{t+1} - \theta_t\|^2 \tag{55}$$

$$= \mathcal{J}_t(\theta_t) - \alpha \|\mathbf{d}_t\|^2 + \frac{\mathbf{G}(1 + \lambda_t)\alpha_t^2}{2} \|\mathbf{d}_t\|^2 \tag{56}$$

$$= \mathcal{J}_t(\theta_t) - \alpha \left(1 - \frac{\alpha \mathbf{G}(1 + \lambda_t)}{2}\right) \|\mathbf{d}_t\|^2. \tag{57}$$

Since $\lambda_t$ is bounded (let $\lambda_t \leq D$) and we choose a sufficiently small step size $\alpha < \frac{2}{\mathbf{G}(1+D)}$, the coefficient $\left(1 - \frac{\alpha \mathbf{G}(1+\lambda_t)}{2}\right)$ is positive. Let this coefficient be $c > 0$. Rearranging the terms, we get:

$$c\alpha \|\mathbf{d}_t\|^2 \leq \mathcal{J}_t(\theta_t) - \mathcal{J}_t(\theta_{t+1}). \tag{58}$$

*Table 6.* Recommended $\varepsilon$ values used for different settings in our experiments

| EXPERIMENT / SETTING | RECOMMENDED $\varepsilon$ | RATIONALE |
|---|---|---|
| Nudity, Violence | $\varepsilon = 1 \times 10^{-3}$ | Sensitive concepts, require tight preservation |
| Artistic Style | $\varepsilon = 1 \times 10^{-3}$ | Global style is easy to remove, require tight preservation |
| Entity | $\varepsilon = 1 \times 10^{-2}$ | Concrete objects harder to disentangle |
| Abstract Concepts | $\varepsilon = 1 \times 10^{-2}$ | Abstract priors harder to disentangle |
| Celebrity Identity | $\varepsilon = 1 \times 10^{-2}$ | Identity preservation more robust to slack |
| Downstream editability / utility (post-erase LoRA use) | $\varepsilon = 5 \times 10^{-3}$ | Maintain editability & style generalization |

However, unlike standard SGD, our objective $\mathcal{J}_t$ changes at each step because $\lambda_t$ updates to $\lambda_{t+1}$. We decompose the total variation:

$$\mathcal{J}_{t+1}(\theta_{t+1}) - \mathcal{J}_t(\theta_t) = \underbrace{\mathcal{J}_{t+1}(\theta_{t+1}) - \mathcal{J}_t(\theta_{t+1})}_{\text{due to } \lambda \text{ update}} + \underbrace{\mathcal{J}_t(\theta_{t+1}) - \mathcal{J}_t(\theta_t)}_{\text{due to } \theta \text{ update}} \qquad (59)$$

$$= (\lambda_{t+1} - \lambda_t)\mathcal{L}_{pr}(\theta_{t+1}) + (\mathcal{J}_t(\theta_{t+1}) - \mathcal{J}_t(\theta_t)). \qquad (60)$$

Summing from $t = 1$ to $T$:

$$\sum_{t=1}^{T} c\alpha \|\mathbf{d}_t\|^2 \leq \mathcal{J}_1(\theta_1) - \mathcal{J}_{T+1}(\theta_{T+1}) + \sum_{t=1}^{T}(\lambda_{t+1} - \lambda_t)\mathcal{L}_{pr}(\theta_{t+1}). \qquad (61)$$

Standard telescope sum analysis combined with the bounded variation of $\lambda$ (from Theorem C.5) ensures the right side is bounded by a constant $C_2$. Therefore, $\sum_{t=1}^{T} \alpha_t \|\mathbf{d}_t\|^2 \leq C_2 < \infty$. Given that $\sum \alpha \to \infty$ (for standard decay rates), it implies that $\liminf_{t\to\infty} \|\mathbf{d}_t\|^2$ must be 0. Specifically, for standard choices of $\alpha$, we have:

$$\min_{t=1,\ldots,T} \|\mathbf{d}_t\|^2 \leq \frac{C_2}{\sum_{t=1}^{T}\alpha} \xrightarrow{T\to\infty} 0. \qquad (62)$$

This proves that the algorithm converges to a point where the update direction $\mathbf{d}_t$ vanishes. Recall that $\mathbf{d}_t$ is the gradient of the Lagrangian. $\mathbf{d}_t = 0$ implies the KKT condition for stationarity is satisfied for the constrained optimization problem. $\square$

**Remark on Hyperparameters.** The theoretical analysis above assumes diminishing step sizes $\alpha, \beta$ to guarantee asymptotic convergence. In our practical implementation (Section 5.1), we adopt fixed hyperparameters ($\alpha = 0.001$, $\beta = 0.1$) for finite-step training. As indicated by Theorem C.5, the stationary error is bounded by terms related to the step sizes. A small, fixed step size effectively maintains the system within a small neighborhood of the optimal Pareto front, balancing convergence speed and stability in the non-convex landscape of diffusion models.

# D. Additional Implementation Details and Results

## D.1. Hyperparameter settings for $\varepsilon$ across experiments

The tolerance parameter $\varepsilon$ controls how tightly the preservation constraint is enforced and directly determines the trade-off between erasing the target concept and maintaining the model's overall utility. Small values of $\varepsilon$ restrict the update direction and prioritize preservation, which is preferable for fragile concepts or datasets with high semantic entanglement. Larger values relax the constraint and enable more aggressive erasure at the cost of controlled performance drift. In all our experiments we find that $\varepsilon$ is robust within a moderate range (typically $10^{-3}$–$10^{-1}$), and does not require fine-grained tuning. We summarize the $\varepsilon$ values used for different experimental settings in Table 6.

## D.2. Complete list of Entity, Artistic Style, Abstraction

To test the generalization and effectiveness of our Z-Erase, we build a concept list at three levels: from the concrete objects to the global artistic style and abstract concepts. The full list used in our experiments is presented in Table 7.

Due to space limits, we provide the main metrics in the main text. Here, we present the full results of miscellaneousness erasure in Table 8.

*Table 7.* Complete list of concepts of Entity, Artistic Style, and Abstraction used in our experiment.

| Category | # Number | Prompt template | Concepts |
|---|---|---|---|
| ENTITY | 10 | 'A photo of [*Entity*]' | '*Church*', '*Soccer*', '*Car*', '*Airplane*', '*Tower*', '*Pencil*', '*Pizza*', '*Shoes*', '*Cat*', '*Clownfish*' |
| ARTISTIC STYLE | 10 | 'An art in the style of [*Artistic Style*]' | '*Pablo Picasso*', '*Salvador Dali*', '*Claude Monet*', '*Vincent Van Gogh*', '*Rembrandt van Rijn*', '*Frida Kahlo*', '*Edvard Munch*', '*Leonardo da Vinci*', '*Nicholas Roerich*', '*Gustave Dore*' |
| ABSTRACTION | 10 | 'A close scene of [*Abstraction*]' | '*Shake Hand*', '*Kiss*', '*Hug*', '*Time*', '*Explosion*', '*Green*', '*Origami*', '*Fried*', '*Amidst*', '*Sad*' |

*Table 8.* **Evaluation on erasing the specific category**: **Entity** (*e.g.,* church), **Artistic Style** (*e.g.,* Claude Monet) and **Abstraction** (*e.g.,* color). CLIP classification accuracy are reported for each category on the erased category itself ($\text{ACC}_e$, efficacy), and the remaining unaffected categories ($\text{ACC}_{ir}$, specificity). The combined performance is summarized by $H_a = \text{ACC}_{ir} - \text{ACC}_e$, where higher values indicate a better balance between erasure and preservation.

| METHOD | ENTITY | | | ARTISTIC STYLE | | | ABSTRACTION | | |
|---|---|---|---|---|---|---|---|---|---|
| | $\text{ACC}_e \downarrow$ | $\text{ACC}_{ir} \uparrow$ | $H_a \uparrow$ | $\text{ACC}_e \downarrow$ | $\text{ACC}_{ir} \uparrow$ | $H_a \uparrow$ | $\text{ACC}_e \downarrow$ | $\text{ACC}_{ir} \uparrow$ | $H_a \uparrow$ |
| AC (Model-Based) | 24.6 | 26.6 | 2.0 | 28.6 | 24.4 | -4.2 | 26.9 | 25.8 | -1.1 |
| AC (Noise-Based) | 25.5 | 26.4 | 0.9 | 29.3 | 23.8 | -5.5 | 27.3 | 26.2 | -1.1 |
| ESD-x | 23.3 | 28.6 | 5.3 | 28.5 | 24.7 | -3.8 | 26.5 | 25.7 | -0.8 |
| ESD-u | 21.8 | 27.8 | 6.0 | 27.8 | 24.5 | -3.3 | 25.7 | 25.3 | -0.4 |
| MACE | 23.2 | 27.5 | 4.3 | 26.6 | 25.8 | -0.8 | 25.1 | 28.5 | 3.4 |
| UCE | 20.1 | 20.6 | 0.5 | **21.7** | 19.9 | -1.8 | **19.7** | 20.3 | 0.6 |
| EraseAnything | 24.1 | 27.6 | 3.5 | 25.9 | 26.1 | 0.2 | 25.6 | **29.0** | 3.4 |
| DiT-Localization | **17.5** | 28.7 | **11.2** | 31.2 | 23.3 | -7.9 | 28.6 | 25.8 | -2.8 |
| Ours | 23.1 | **28.9** | 7.4 | 26.1 | **26.9** | **0.8** | 24.3 | **29.0** | **4.7** |
| Z-Image Turbo | 24.7 | 32.1 | - | 33.6 | 27.6 | - | 30.5 | 29.2 | - |

## D.3. Complete list of Celebrities

The celebrities used in our experiments are illustrated in Table 9. It's noteworthy that not arbitrary celebrities can be faithfully synthesized by Z-Image Turbo, after manually comparing the synthesized famous people with its prompt and add some comic characters, we keep 50 for each group.

## E. User Study

A common limitation in prior work on concept erasure and attack is the reliance on pretrained detectors or classifiers as the sole evaluation criterion. While such tools offer scalability, they are often unreliable: detectors may miss subtle instances of a concept (false negatives), mistakenly flag benign patterns (false positives), or fail to distinguish between high-quality erasure and degraded artifacts. This creates a gap between machine-detected presence of a concept and human-perceived restoration.

To overcome these shortcomings, we conduct a user study, involving 30 non-artist participants, each providing an average of 50 responses. Adhering to Z-Image's comprehensive evaluative criteria for T2I models, our user study incorporates three foundational metrics: **Image Quality**, **Prompt Adherence**, and **Output Diversity**. Together, these criteria form the evaluative baseline for benchmarking generative performance. In the specific context of concept erasure, where the objective is to suppress the target concept while allowing unrelated concepts to be faithfully rendered, this baseline is augmented with two task-oriented metrics: **Erasing Cleanliness** and **Irrelevant Preservation**.

*Table 9.* Complete list of celebrities used in our experiment.

| Category | # Number | Mapping Concept | Celebrities |
|---|---|---|---|
| Erasure Group | 50 | 'A person' | 'Adele', 'Albert Camus', 'Angelina Jolie', 'Arnold Schwarzenegger', 'Audrey Hepburn', 'Barack Obama', 'Beyoncé', 'Brad Pitt', 'Bruce Lee', 'Chris Evans', 'Christiano Ronaldo', 'David Beckham', 'Dr Dre', 'Drake', 'Elizabeth Taylor', 'Eminem', 'Elon Musk', 'Emma Watson', 'Frida Kahlo', 'Hugh Jackman', 'Hillary Clinton', 'Isaac Newton', 'Jay-Z', 'Justin Bieber', 'John Lennon', 'Keanu Reeves', 'Leonardo Dicaprio', 'Mariah Carey', 'Madonna', 'Marlon Brando', 'Mahatma Gandhi', 'Mark Zuckerberg', 'Michael Jordan', 'Muhammad Ali', 'Nancy Pelosi','Neil Armstrong', 'Nelson Mandela', 'Oprah Winfrey', 'Rihanna', 'Roger Federer', 'Robert De Niro', 'Ryan Gosling', 'Scarlett Johansson', 'Stan Lee', 'Tiger Woods', 'Timothee Chalamet', 'Taylor Swift', 'Tom Hardy', 'William Shakespeare', 'Zac Efron' |
| Retention Group | 50 | - | 'Angela Merkel', 'Albert Einstein', 'Al Pacino', 'Batman', 'Babe Ruth Jr', 'Ben Affleck', 'Bette Midler', 'Benedict Cumberbatch', 'Bruce Willis', 'Bruno Mars', 'Donald Trump', 'Doraemon', 'Denzel Washington', 'Ed Sheeran', 'Emmanuel Macron', 'Elvis Presley', 'Gal Gadot', 'George Clooney', 'Goku','Jake Gyllenhaal', 'Johnny Depp', 'Karl Marx', 'Kanye West', 'Kim Jong Un', 'Kim Kardashian', 'Kung Fu Panda', 'Lionel Messi', 'Lady Gaga', 'Martin Luther King Jr.', 'Matthew McConaughey', 'Morgan Freeman', 'Monkey D. Luffy', 'Michael Jackson', 'Michael Fassbender', 'Marilyn Monroe', 'Naruto Uzumaki', 'Nicolas Cage', 'Nikola Tesla', 'Optimus Prime', 'Robert Downey Jr.', 'Saitama', 'Serena Williams', 'Snow White', 'Superman', 'The Hulk', 'Tom Cruise', 'Vladimir Putin', 'Warren Buffett', 'Will Smith', 'Wonderwoman' |

Erasing Cleanliness quantifies the effectiveness of the erasure mechanism, assessing whether the target concept is not only removed but eliminated without residual traces or indirect stylistic leakage. Irrelevant Preservation, in contrast, measures the system's capability to retain semantic fidelity for concepts outside the erasure domain, ensuring that composition, contextual cues, and stylistic attributes remain coherent and unaffected, and the image is high-quality without artifacts.

Fig. 10 and 11 illustrate the user study interface, which is deliberately designed to support smooth participant interaction and minimize cognitive overhead. During the evaluation, participants are shown image sets containing either 4 or 3 samples produced by multiple anonymized methods. They are asked to rate each method across the five aforementioned metrics using a consistent scoring protocol. The collected ratings are subsequently aggregated and visualized as a pentagonal performance profile (Fig. 6 in the main paper), providing an intuitive and comparative view of each method's strengths and weaknesses.

This representation offers both researchers and practitioners a structured and interpretable summary of model capabilities, enabling fine-grained insights into the trade-offs inherent in concept erasure and guiding future methodological refinements in T2I generation.

## F. Multi-Concept Erasure

In this section, we elaborate on the strategy employed for erasing multiple concepts simultaneously using Z-Erase. While the primary experiments in the main text focus on single-concept erasure to rigorously validate our Lagrangian-Guided Adaptive Erasure Modulation, practical safety alignment often requires removing multiple undesirable concepts at once.

**Linear Synergy of Concept-Erased LoRAs.** Our approach to multi-concept erasure leverages the modularity of Low-Rank Adaptations (LoRAs). Since Z-Erase confines the parameter updates to a specific subspace via Stream Disentangled Concept Erasure Framework, the optimization trajectories for different concepts remain relatively orthogonal in the parameter space. This structural decoupling allows us to merge independent erasure modules into a cohesive single entity without retraining from scratch.

# User Study

This study evaluates AI generated images from different concept erasure methods, each time showing 4 results under the same prompt and seed. You need to judge the results with two metrics: **Erasing Cleanliness** and **Irrelevant Preservation**.

Please rate the images from 1 to 5, where **1** indicates the **lowest** quality and **5** represents the **highest** quality. The column in RED is designated for evaluating **Erasing Cleanliness** of the images, which refer to the effectiveness of the methods to erase the target concept. The adjacent column in BLUE is for **Irrelevant Preservation**, which measures how well the methods can generate images that are not intended for erasure.

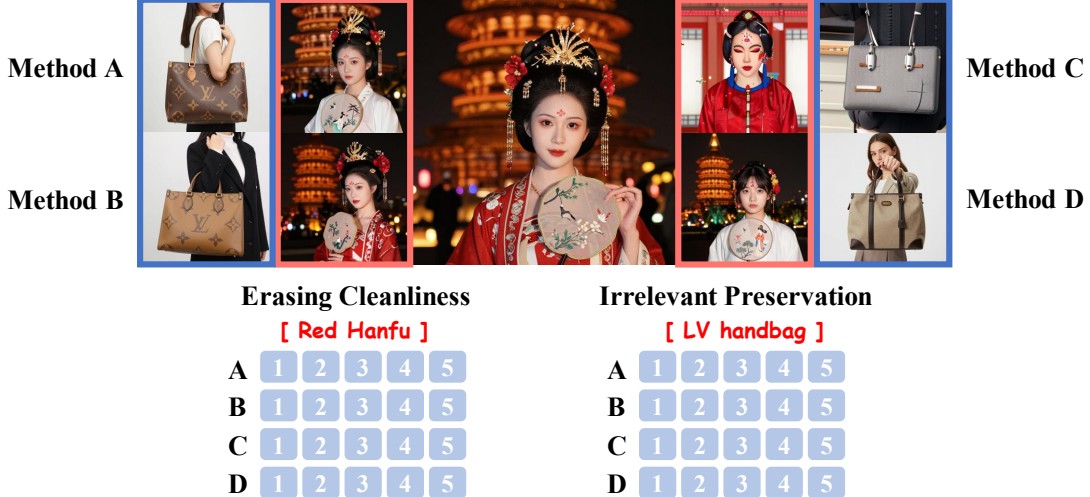

Figure 10. **User study interface on Erasing Cleanliness and Irrelevant Preservation.**

# User Study

This study evaluates AI generated images from different concept erasure methods, each time showing 3 results under the same prompt and seed. You need to judge the results with three metrics: **Image Quality**, **Prompt Adherence** and **Output Diversity**.

Please rate the images from 1 to 5, where **1** indicates the **lowest** quality and **5** represents the **highest** quality. Prompts are all sampled from PromptHero.

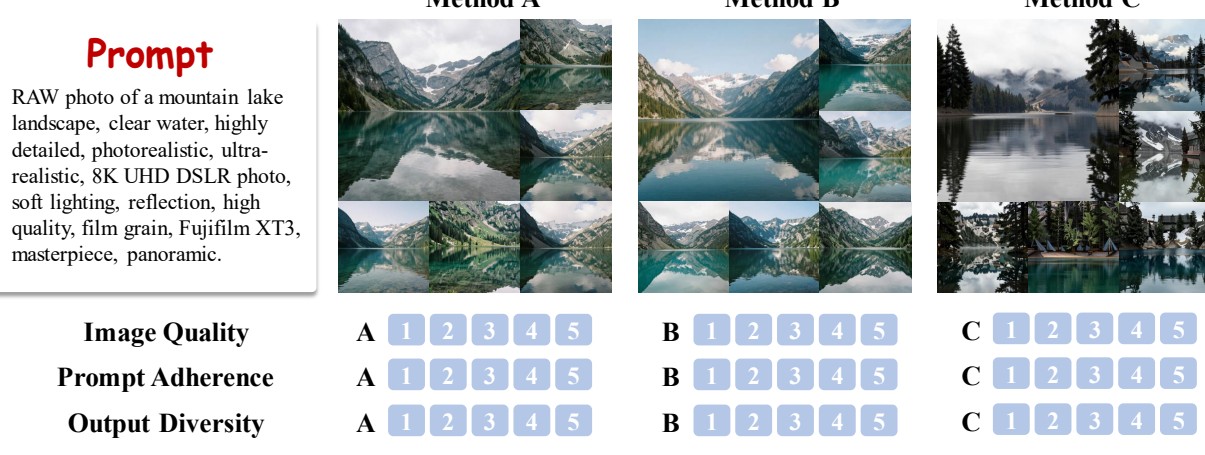

Figure 11. **User study interface on Image Quality, Prompt Adherence and Output Diversity.**

To unify multiple erasure objectives, we employ a linear interpolation strategy in the parameter space. Let $\{\Delta\theta_1, \Delta\theta_2, \ldots, \Delta\theta_N\}$ represent a set of $N$ independently trained LoRA modules, each targeting a specific concept $c_i \in \{c_1, \ldots, c_N\}$. The combined multi-concept LoRA, denoted as $\Delta\theta_{\text{mul}}$, is computed as a weighted average:

$$\Delta\theta_{\text{mul}} = \sum_{i=1}^{N} w_i \Delta\theta_i, \tag{63}$$

*Table 10.* **Additional quantitative results on HunyuanImage-3.0 model.**

| METHOD | NUDITY | | | ENTITY | | | ARTISTIC STYLE | | | ABSTRACTION | | |
|---|---|---|---|---|---|---|---|---|---|---|---|---|
| | Total ↓ | FID ↓ | CLIP ↑ | $\text{ACC}_e$ ↓ | $\text{ACC}_{ir}$ ↑ | $\text{H}_a$ ↑ | $\text{ACC}_e$ ↓ | $\text{ACC}_{ir}$ ↑ | $\text{H}_a$ ↑ | $\text{ACC}_e$ ↓ | $\text{ACC}_{ir}$ ↑ | $\text{H}_a$ ↑ |
| DiT-Localization (Zarei et al., 2025) | 306 | 36.7 | 26.3 | **16.8** | 23.4 | **6.6** | 30.7 | 23.1 | -7.6 | 27.9 | 26.4 | -1.5 |
| EraseAnything (Gao et al., 2025a) | 231 | 24.0 | 29.9 | 24.6 | 27.5 | 2.9 | **25.6** | 26.4 | **0.8** | 25.3 | **28.9** | 3.6 |
| Ours | **145** | **23.6** | **30.8** | 24.3 | **27.9** | 3.6 | 26.2 | **27.0** | **0.8** | 24.9 | 28.8 | **3.9** |
| HunyuanImage-3.0 | 628 | 23.1 | 31.2 | 28.6 | 31.5 | - | 32.1 | 28.3 | - | 29.6 | 29.8 | - |

where $w_i$ represents the merging weight for the $i$-th concept. We adopt a normalized weight blending strategy where $w_i = \frac{1}{N}$. This normalization is crucial; it ensures that the magnitude of the combined perturbation remains within the safe optimization manifold established by our preservation constraints. Simply summing the weights (i.e., $\sum w_i > 1$) often leads to concept "over-erasure" or catastrophic forgetting of general knowledge, as the aggregated parameter shift pushes the model too far from its pre-trained state.

By averaging the weights, Z-Erase effectively constructs a superposition of distinct erasure directions. As demonstrated in Fig. 7, this simple yet effective strategy successfully removes multiple target concepts simultaneously. The generated images confirm that the model ceases to produce the target concepts while maintaining high fidelity for the remaining unrelated context, validating the linearity and composability of the LoRA modules learned by Z-Erase.

# G. Additional Results and Analysis

## G.1. Evaluation on HunyuanImage-3.0

Due to space constraints in the main text, our primary evaluation focused on Z-Image Turbo. To further validate the versatility and robustness of Z-Erase across different single-stream architectures, we provide additional experimental results on HunyuanImage-3.0, another leading open-source model in this domain. We conduct tests using the I2P dataset (Schramowski et al., 2023) for nudity erasure and our curated Entity, Artistic Style, and Abstraction datasets (Table 7) for concept erasure, maintaining the exact same hyperparameter configurations as detailed in the main text.

As shown in Table 10, Z-Erase demonstrates consistent superiority on the HunyuanImage-3.0 model. Our method achieves the lowest total nudity detection count while maintaining FID and CLIP scores highly comparable to the original model, significantly outperforming baselines like EraseAnything and DiT-Localization. In miscellaneousness concept erasure tasks (Entity, Style, Abstraction), Z-Erase consistently attains high balance scores ($\text{H}_a$), indicating that we do not sacrifice image quality or irrelevant preservation for erasure success. Notably, our advantage becomes more pronounced when dealing with abstract and global concepts. Since these concepts are often entangled deeply within the high-level semantic layers of single-stream models, prior methods struggle to isolate them without damaging the model's generation capability. In contrast, Z-Erase effectively navigates this complexity. Visual comparisons on HunyuanImage-3.0 are provided in Fig. 12 and Fig. 13, showing that Z-Erase cleanly removes target concepts without introducing the artifacts or semantic drifts.

## G.2. Detailed Analysis of Baselines and Failure Modes

In this section, we provide a deeper analysis of why existing methods struggle in the single-stream paradigm, even after being stabilized by our proposed mechanism.

It is important to note that all baselines evaluated in this work were adapted using our **Stream Disentangled Concept Erasure Framework**. Without this structural decoupling, naive fine-tuning will lead to immediate generation collapse (as shown in Fig. 1). However, even with this safe optimization subspace, prior methods frequently fall into one of two failure modes: **under-erasure** or **over-erasure**.

1. **Under-erasure**: The target concept is not fully removed (*e.g.*, subtle traces remain) or reappears under slight prompt perturbations or adversarial attacks.

2. **Over-erasure**: The model's general utility is damaged, leading to severe visual artifacts, "fried" textures, or the inability to generate unrelated concepts.

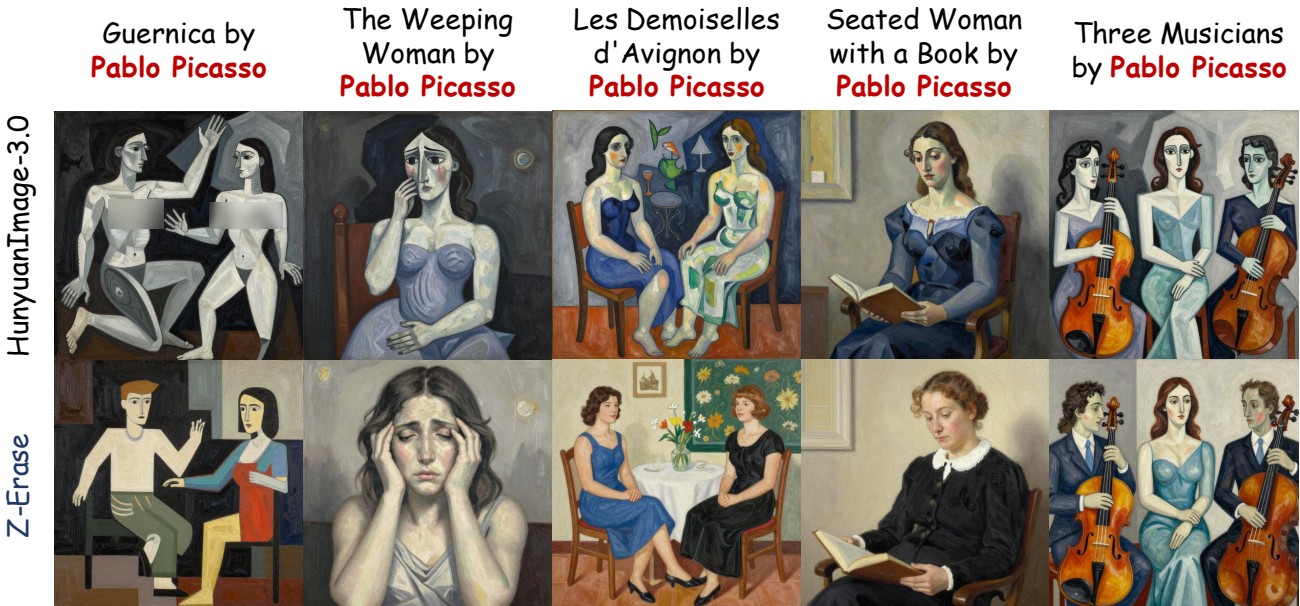

*Figure 12.* **Visualizations on erasing the concept "Pablo Picasso"**. With the retention prompt kept as "painting" in LoRA training, our method preserves the overall painting modality while effectively removing Picasso's signature abstract-geometric style. The generated results remain high-quality and faithful, exhibiting minimal semantic or compositional shift.

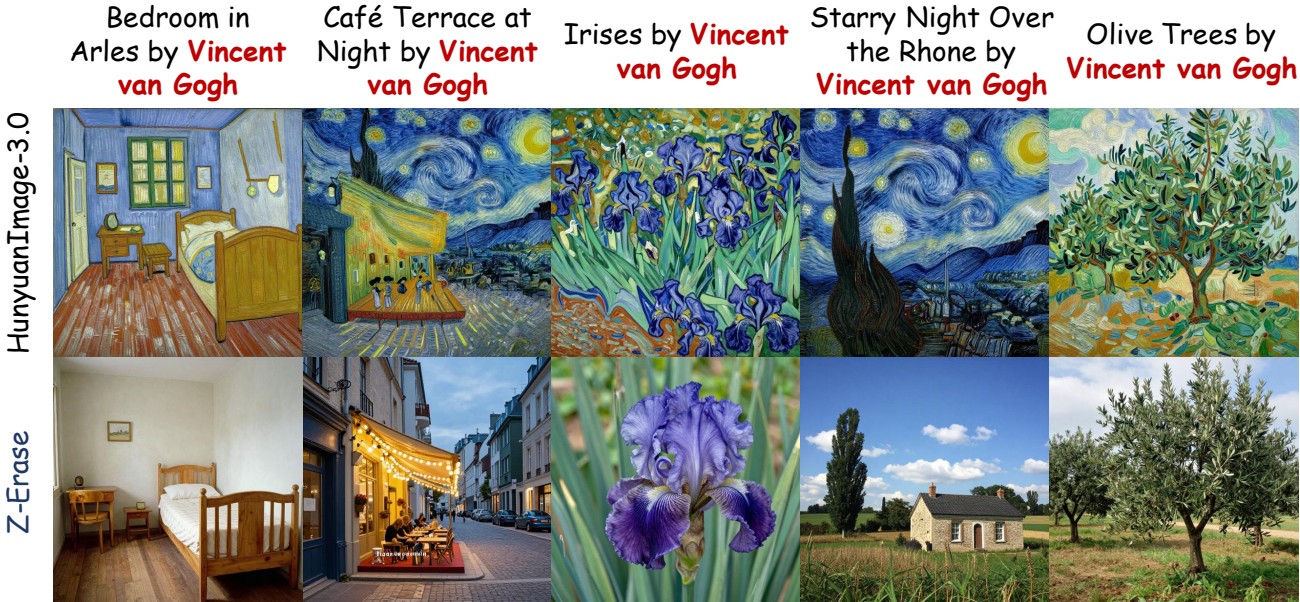

*Figure 13.* **Visualizations on erasing the concept "Vincent van Gogh"**. With the retention prompt kept as "realistic" in LoRA training, our method generates realistic scenarios while effectively removing van Gogh's signature abstract style. The generated results remain high-quality and faithful, exhibiting minimal semantic or compositional shift.

This binary outcome is exacerbated by the highly entangled nature of single-stream transformers. Traditional optimization objectives lack the granularity to navigate this sensitive landscape. We analyze two recent representative strong baselines to illustrate this:

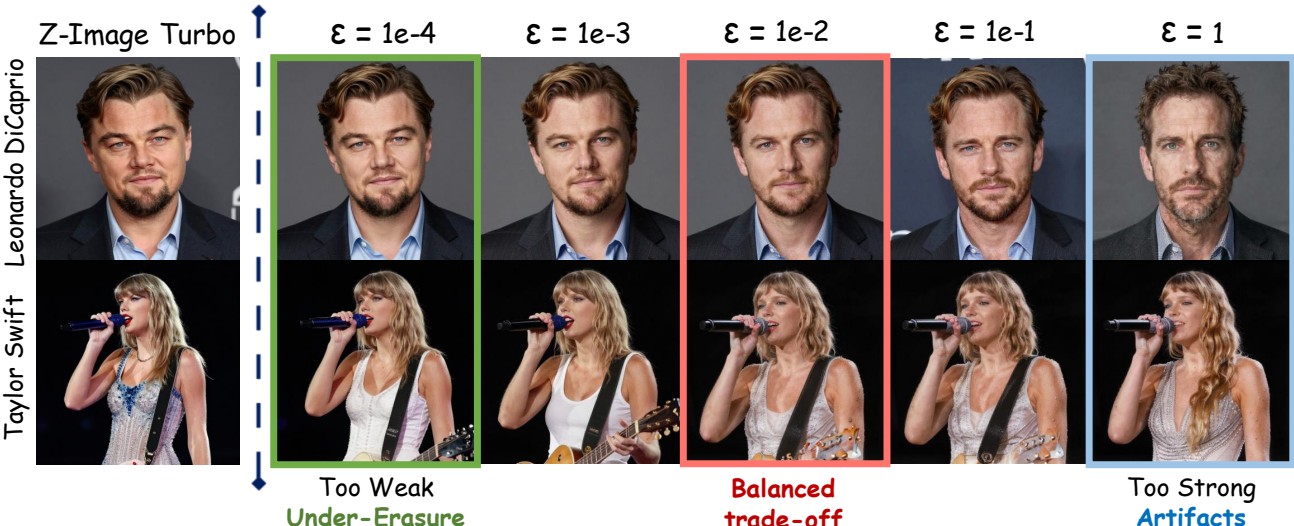

*Figure 14.* **Visualizations of ablation study on $\varepsilon$.**

**Analysis of EraseAnything** (Gao et al., 2025a). EraseAnything employs a bi-level optimization strategy to find a balance between erasing and preserving. While theoretically sound for dual-stream models (like Flux), we find its granularity is too coarse for single-stream architectures. It struggles to find the optimal trade-off point on the Pareto front. In practice, it often settles for a conservative solution (leading to under-erasure) or pushes too hard to erase, causing semantic drift in the preservation set. It lacks a dynamic "brake" mechanism to stop erasure precisely when utility starts to degrade.

**Analysis of DiT-Localization** (Zarei et al., 2025). DiT-Localization operates by calculating attention scores for the target concept across layers and then replacing the text embedding with a null token in the Top-K most responsive layers. As seen in Table 10, this method is indeed powerful at erasure (low nudity count). However, it suffers significantly in utility preservation (high $ACC_{ir}$, lower $H_a$). In single-stream models, layers do not have discrete roles (*e.g.*, "just style" or "just object"). An attention layer responsive to "nudity" might also be responsible for generating "skin texture" or "human anatomy" in general. By bluntly nulling out these layers, DiT-Localization acts like a sledgehammer: it removes the target but often breaks the generation of unrelated concepts that rely on those same shared layers.

These observations underscore the necessity of our **Lagrangian-Guided Adaptive Erasure Modulation**. Instead of a static optimization target or a heuristic layer-pruning strategy, Z-Erase dynamically negotiates the trade-off at every step. This allows it to act as a scalpel, surgically removing the target concept deep within the entangled weights while strictly respecting the preservation boundary.

### G.3. Additional ablation study results

In this section, we provide further insights into the core components of Z-Erase through visual ablations. These qualitative results serve to reinforce the quantitative findings discussed in the main text, offering a clearer window into how our method navigates the complex optimization landscape of single-stream models.

**Visualization of the Preservation Constraint $\varepsilon$.** While our main text quantifies the impact of $\varepsilon$ on the I2P dataset, visualizing its effect on specific identity erasure offers a more intuitive understanding of the erasure-preservation trade-off. In Fig. 14, we focus on two distinct celebrity concepts: *Leonardo DiCaprio* and *Taylor Swift*. As we traverse the value of $\varepsilon$ from $1e-4$ to 1, we observe a monotonic and controllable shift in the model's behavior. At extremely tight constraints, the model prioritize preservation too heavily, resulting in under-erasure where the identities remain virtually unchanged. Conversely, loosening the constraint too much leads to aggressive erasure that introduces severe visual artifacts—distorting facial features and degrading overall image quality. The "sweet spot" (highlighted in red) represents a balanced state where the target identity is successfully removed while high-quality, photorealistic generation is maintained. This capability to monotonically tune the erasure intensity is a distinct advantage of Z-Erase, allowing users to define their own safety-utility boundaries.

**Necessity of Stream Disentangled Concept Erasure Framework.** Finding the correct parameters to tune in a single-stream

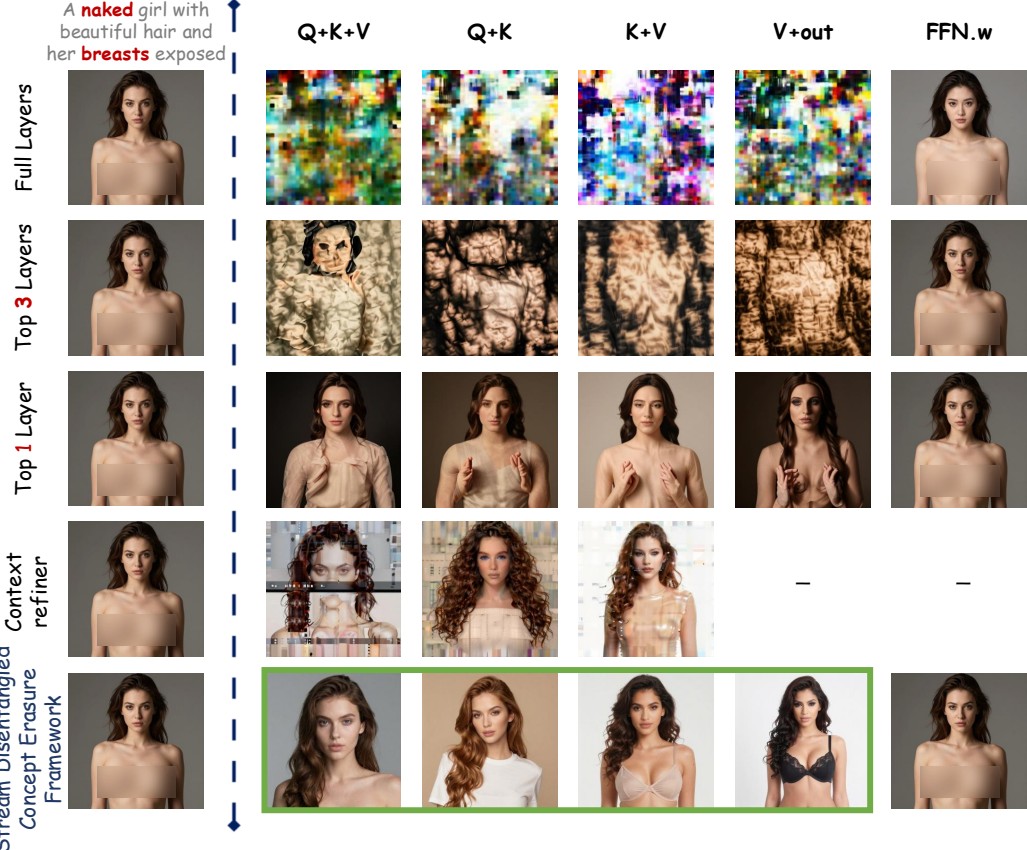

*Figure 15.* **Visual ablation of different fine-tuning configurations.** We compare various parameter update strategies to demonstrate the necessity of our method. "Top K" and "Top 1" layers are selected based on the highest attention scores for the target concept, following the logic of DiT-Localization (Zarei et al., 2025). The results reveal critical insights: fine-tuning the FFN weights has negligible impact on erasure; meanwhile, any standard LoRA fine-tuning on shared projections (even when restricted to a single layer) leads to immediate generation collapse and severe visual artifacts. Only our *Stream Disentangled Concept Erasure Framework* (bottom row) achieves high-quality erasure while preserving the structural integrity of the image. Therefore, our *Stream Disentangled Concept Erasure Framework* is a prerequisite for stable erasure in pure single-stream models.

architecture is not a straightforward process. The high entanglement between text and image processing means that naive intervention strategies almost invariably lead to generation collapse. To illustrate this difficulty and justify our comprehensive design, we demonstrate the failure modes of various alternative fine-tuning configurations in Fig. 15.

We experimented with numerous combinations, including tuning different subsets of projection weights (Query, Key, Value) and restricting updates to specific layers (Top-1, Top-3, Full Layers) based on attention scores derived from DiT-Localization (Zarei et al., 2025). The results are stark: Directly fine-tuning shared projections (regardless of the combination $(Q + K + V, Q + K, etc.)$ or the number of layers, or context refiner) results in catastrophic noise. Even modifying a single layer (Top-1 Layer) is sufficient to destroy the image structure and lead to strong visual artifacts. Interestingly, we find that fine-tuning the Feed-Forward Network (FFN) had almost no impact on erasure, suggesting that semantic binding is primarily encoded in the attention mechanism. Ultimately, only our proposed *Stream Disentangled Concept Erasure Framework*, which structurally isolates the parameter updates to the textual pathway, enables effective concept erasure without compromising visual integrity. This confirms that our architectural intervention is not merely an option, but a prerequisite for stable erasure in single-stream diffusion transformers.

### G.4. Additional Visualizations

Here, we provide more visualizations compared with state-of-the-art erasure methods adapted by our *Stream Disentangled Concept Erasure Framework*. As shown in Figure 16, our Z-Erase removes a wide range of concepts, from concrete to abstraction. Moreover, to assess the potential influence of integrating fine-tuned LoRAs into the original model, as

depicted in Figure 17, it can be observed that incorporating fine-tuned LoRAs for diverse concepts, *i.e. Celebrity*: **Batman, Spiderman, Cristiano Ronaldo, Hulk, Ironman**. *Entity*: **Tiger, Cat, Basketball, Statue of Liberty, The Great Wall** and *Art*: **Van Gogh, Edvard Munch, Leonardo da Vinci, Johannes Vermeer, Rembrandt**, does not adversely affect the original image synthesis capabilities. All above-mentioned concepts are depicted sequentially from left to right.

## H. Limitations

Compared to naive fine-tuning, our adaptive modulation introduces a slight computational cost. This is due to the calculation of the preservation loss gradient ($\nabla \mathcal{L}_{pr}$) required for the constraint mechanism. Specifically, on a single H20 GPU, standard naive fine-tuning takes approximately 20 minutes per concept. In comparison, Z-Erase requires about 35 minutes to converge. We believe this extra time is a worthy trade-off for the significant stability and quality improvements achieved.

## I. Ethics Statement

The rapid evolution of generative AI has brought us to the era of *Single-Stream Diffusion Transformers*, a unified architecture that promises unprecedented generation quality and efficiency. However, this unification of text and image processing also introduces new risks. The deep entanglement of parameters makes these foundational models harder to control, posing new challenges for safe deployment.

Z-Erase is developed in response to this emerging challenge. Our work is guided by the following ethical principles:

1. **Proactive safety alignment**: We believe safety should not be an afterthought. As architectures evolve from U-Net to Single-Stream Transformers, safety mechanisms must evolve with them. Z-Erase ensures that the capability to remove harmful content (NSFW, violence) remains effective in this new paradigm.

2. **Copyright protection**: Generative models often inadvertently memorize copyrighted materials. By enabling precise concept erasure, we provide model creators and community users with a tool to respect intellectual property and remove artistic styles or characters upon request.

3. **Safety without compromise**: A truly safe foundation model should not sacrifice its capabilities. Unlike blunt erasure methods that degrade image quality or irrelevant concepts, Z-Erase is designed to strictly preserve the model's original powerful performance on unrelated tasks. We aim to build both secure and highly capable models, ensuring the next generation of T2I models serves society effectively.

In summary, Z-Erase represents a step towards *Responsible AI*. We hope this work inspires further research into the interpretability and controllability of single-stream architectures, ensuring that the next generation of foundational models serves society safely and ethically.

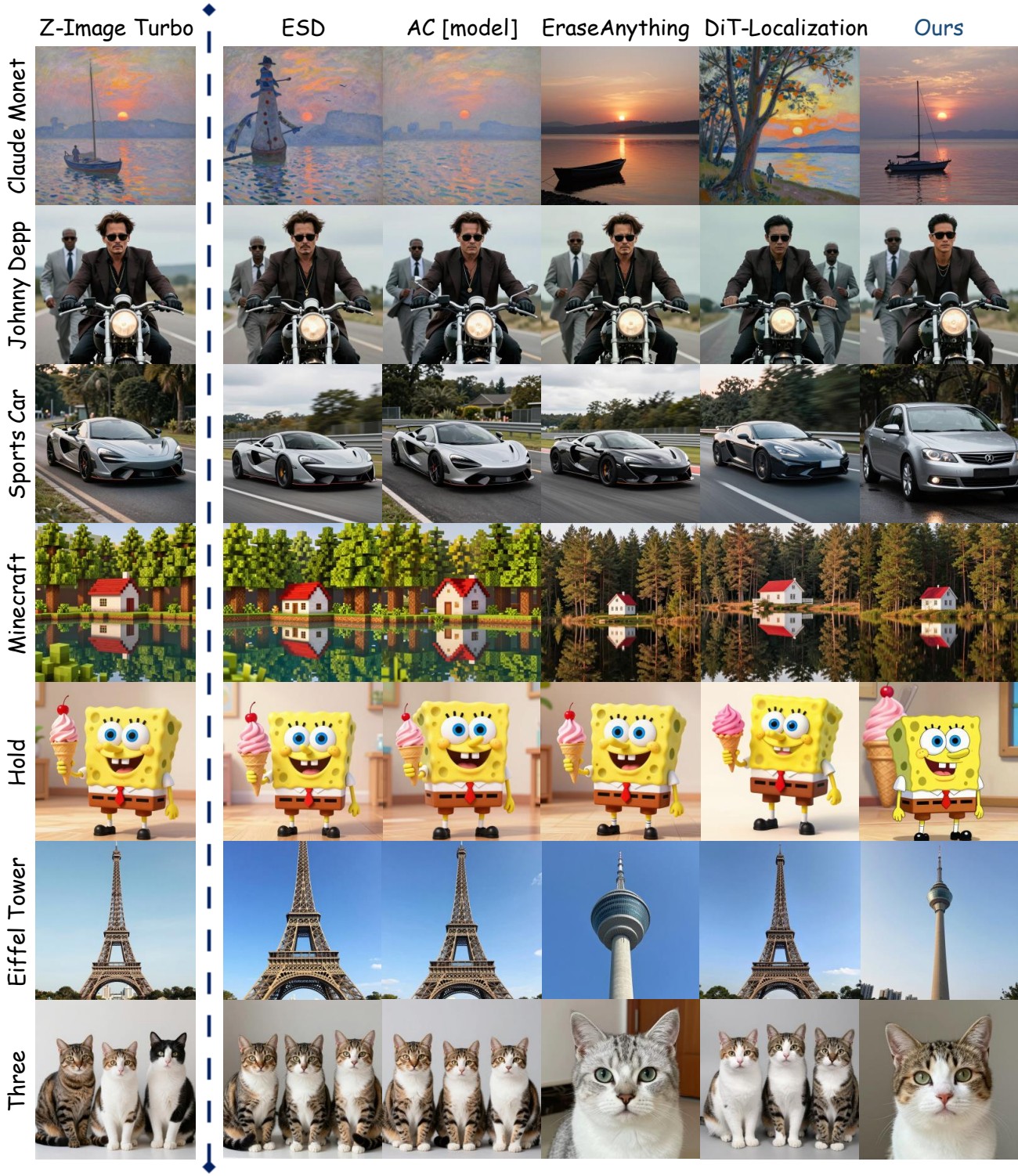

*Figure 16.* **Comparison with mainstream concept erasing methods.** Here, ESD, AC and EraseAnything are adapted with our *Stream Disentangled Concept Erasure Framework* to prevent generation collapse.

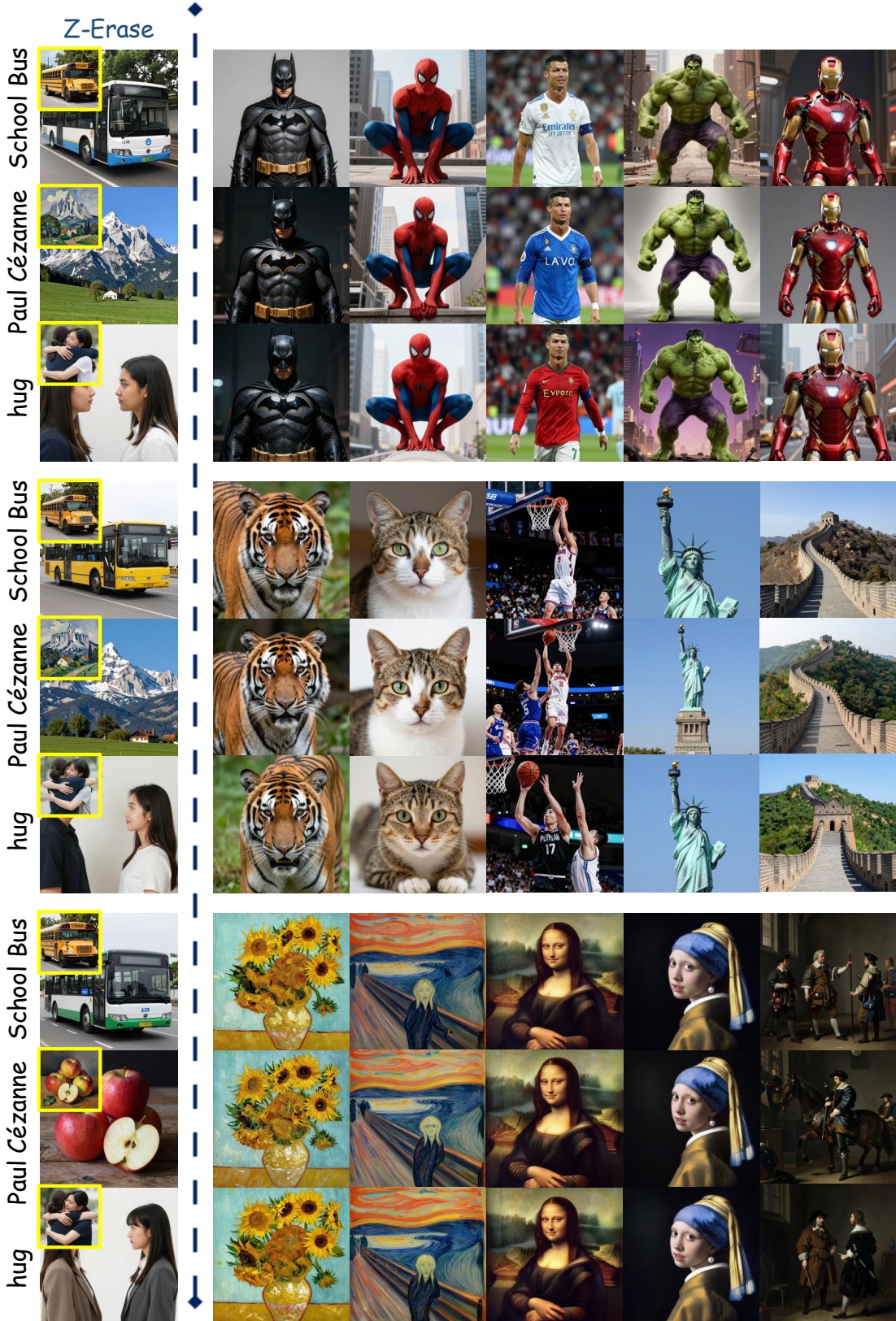

*Figure 17.* **Visualization on LoRA Disentanglement.** The left side of the blue dashed line delineates the original image (yellow box) and the erased image by our Z-Erase. The right side illustrates the result on unrelated concepts upon incorporating the LoRA associated with the erased concept. Top rows: *Celebrity*; Mid rows: *Entity*; Last rows: *Art*.

