# OpenReview forum: "Z-Erase: Enabling Concept Erasure in Single Stream Diffusion Transformers"
_ICML.cc/2026/Conference — ICML 2026 regular_

### Official Review · Reviewer_kpDa · 2026-03-05

**Soundness:** 3
**Presentation:** 3
**Significance:** 2
**Originality:** 3
**Overall Recommendation:** 4
**Confidence:** 4

**Summary:**

The paper identifies why existing weight-edit concept-erasure methods fail on single-stream diffusion transformers. Based on this, it formulates erasure and preservation objectives and optimizes them jointly using a Lagrangian-guided modulation scheme. This approach enables effective concept erasure while largely preserving overall model utility

**Compliance With Llm Reviewing Policy:**

Affirmed.

**Final Justification:**

The authors resolved my main concerns regarding work in the rebuttal by running additional evaluations. To this end I raised my score to Weak Accept as I think that it matches my recommendation best: it's a technically solid paper, though since it's main contribution relies on `Stream Decoupled Framework` that edits how prior unlearning methods work in a architecture of single-stream diffusion transformers, it's unlear to me how big of an impact this work will have on the field.

**Key Questions For Authors:**

* Did you test any representation-level unlearning baselines (e.g., SAE-based or steering-vector approaches)? If not, what prevents applying them in this setting, and do you expect them to fail for the same coupling reasons that break weight-edit approaches?

**Limitations:**

yes

**Strengths And Weaknesses:**

## Strengths
* The paper provides a clear and important finding of limitations of attention-intervention and fine-tuning-based erasure approaches to unlearning
* Figures are clear and well designed.
* The Lagrangian-based optimization is a novel and elegant way to balance erasure and preservation objectives.
* The evaluation is thorough within the class of concept-erasure baselines considered.

## Weaknesses
* The baselines are almost entirely fine-tuning / weight-edit methods. The paper does not test whether other unlearning approaches already transfer to single-stream diffusion transformers without architecture-specific modifications—for example, representation-level interventions such as sparse-autoencoder based [1] or steering-vector methods [2]. Including such comparisons would strengthen the motivation for introducing an architecture-specific unlearning method.

[1] Cywiński, Bartosz, and Kamil Deja. "SAeUron: Interpretable Concept Unlearning in Diffusion Models with Sparse Autoencoders." Forty-second International Conference on Machine Learning.

[2] Gaintseva, Tatiana, et al. "CASteer: Cross-Attention Steering for Controllable Concept Erasure." ICLR 2026

---

> ### Author Rebuttal · Authors · 2026-03-28
>
> Thank you for your insightful feedback! We implemented ``SAeUron`` and ``CASteer`` on Z-Image. They fail for the exact same coupling reasons. Please see the visual results here: https://imgur.com/a/2xJBYUT.
>
> - **SAeUron (SAE-based):** It intervenes on explicit Cross-Attention (CA) features. In single-stream models, explicit CA does not exist. Directly ablating unified representations inevitably distorts shared visual signals, causing strong artifacts. It only optimizes properly when equipped with our Stream Disentangled Concept Erasure Framework.
> - **CASteer (Steering-vector):** It works by adding steering vectors to CA outputs. Without CA layers, forcing steering vectors into the unified stream destroys structural integrity, leading to severe generation collapse. It similarly requires our Stream Disentangled Concept Erasure Framework to function safely.
>
> In conclusion, representation-level interventions cannot transfer out-of-the-box. Introducing an architecture-specific unlearning method for single-stream diffusion transformers is absolutely necessary.
>
> To quantitatively prove this, we evaluated these adaptations on the I2P dataset (same as the settings of ``Table 1``):
>
> | **Method**                                   | **Detected Nudity (Total) ↓** | **FID ↓**                  | **CLIP ↑** |
> | -------------------------------------------- | ----------------------------- | -------------------------- | ---------- |
> | ``Pure`` SAeUron                             | 458                           | 57.12 (*Strong Artifacts*) | 22.30      |
> | SAeUron + ``our Stream Decoupled Framework`` | 385                           | 28.15                      | 29.80      |
> | ``Pure`` CASteer                             | *Collapse*                    | -                          | -          |
> | CASteer + ``our Stream Decoupled Framework`` | 178                           | 34.60                      | 26.85      |
> | **Ours**                                     | **161**                       | **26.46**                  | **31.25**  |
> | Z-Image Turbo (Original)                     | 649                           | 26.33                      | 31.47      |

---

> > ### Author Rebuttal · Reviewer_kpDa · 2026-04-03
> >
> > Thank you for running this additional evaluation of representation-level baselines. I think it further strengthens the evidence that the `Stream Decoupled Framework` proposed by the authors is needed for unlearning methods to work in single-stream diffusion transformers.
> >
> > Since my concerns have been resolved, I raise my score and encourage authors to include the additional evaluation results in the final version of the manuscript.

---

> > > ### Author Response · Authors · 2026-04-03
> > >
> > > Thank you for the feedback! We will include the additional experimental results in the final version.

---

### Official Review · Reviewer_z3eS · 2026-03-11

**Soundness:** 3
**Presentation:** 3
**Significance:** 2
**Originality:** 2
**Overall Recommendation:** 4
**Confidence:** 3

**Summary:**

This paper introduces Z-Erase, the first concept erasure method specifically designed for single-stream diffusion transformers. The authors identify that directly applying prior erasure techniques to single-stream models causes "generation collapse" because textual and visual data share a unified sequence and parameters. To resolve this, the work proposes a Stream Disentangled Concept Erasure Framework that isolates updates to textual hidden states to protect the visual backbone , alongside a Lagrangian-Guided Adaptive Erasure Modulation algorithm to dynamically balance the trade-off between erasing target concepts and preserving model utility. Experimental results across NSFW, celebrity, and style erasure tasks demonstrate that Z-Erase achieves state-of-the-art performance while maintaining high image quality。

**Compliance With Llm Reviewing Policy:**

Affirmed.

**Final Justification:**

The rebuttal has addressed my concern. I change my score to 4.

**Key Questions For Authors:**

Refer to weakness.

**Limitations:**

yes

**Strengths And Weaknesses:**

Strengths:
1. The proposed method is simple and easy to follow.
2. The proposed method achieves excellent performance across various benchmarks and tasks.

Weaknesses:
1. First, the underlying motivation of this work remains unclear to me. If existing concept erasure methods are compatible with hybrid architectures like FLUX (which combine single-stream and dual-stream blocks), why would they fail to generalize to a purely single-stream architecture like Z-Image? Decoupling the processing of visual and textual modalities within a single-stream framework is not inherently restrictive.
2. The presentation of the paper requires significant refinement. For instance, Figure 2 conveys very little informative value. Furthermore, the description of model fine-tuning in Section 4.1 appears unnecessarily convoluted. If my understanding is correct, the approach simply involves applying LoRA to the QKV projection layers corresponding to the text tokens.
3. Regarding the real-world prompt variations illustrated in Figure 3, it is not explicitly clear how the proposed solution specifically addresses or mitigates this particular challenge.

---

> ### Author Rebuttal · Authors · 2026-03-28
>
> Thank you for your kind words and recognition! We address them as follows:
>
> - **(A) Motivation: The Hidden Truth Behind FLUX Baselines**
>
>   Thank you for this critical question! The misunderstanding stems from a hidden implementation detail in existing FLUX baselines: **they completely bypass the single-stream blocks.**
>
>   Reviewing the open-source code for FLUX concept erasure (e.g., EraseAnything [1], MACE [2], Minimalist CE [3]), they strictly confine fine-tuning to the **text-only branches** within the **dual-stream** blocks. They intentionally comment out all shared/single-stream modules:
>
>   ```python
>   # [ICML 2025] EraseAnything: train_flux_lora.py (Line 205)
>   target_modules = [
>       # "attn.to_k",        # Skipped: Image/Shared branch
>       # "attn.to_q",        # Skipped: Image/Shared branch
>       "attn.add_k_proj",    # Tuned: Text branch (Dual-stream only)
>       "attn.add_q_proj",    # Tuned: Text branch (Dual-stream only)
>       ...
>   ]
>   ```
>
>   Why do they avoid the single-stream modules? Because **updating shared weights inherently destroys visual synthesis**. To prove this, we forced standard LoRA onto FLUX's single-stream blocks. As expected, it triggered the exact same catastrophic generation collapse we observed in Z-Image: https://imgur.com/a/mJXHEFs.
>
>   In short: FLUX baselines rely on the architectural luxury of isolated text branches. Pure single-stream models (like Z-Image) do not have this luxury; text and images share parameters entirely. Therefore, our explicit structural decoupling (Stream Disentangled Framework) is an absolute necessity.
>
> - **(B) Presentation and Illustration**
>
>   Thank you for the feedback. We will refine the writing to make it more intuitive. ``Figure 2`` is to empirically visualize that even without explicit cross-attention modules, single-stream self-attention inherently forms stable text-image interactions. This critical finding is the direct inspiration for ``Eq. (6)``, and we will clarify this connection. Furthermore, your understanding is perfectly correct: the implementation effectively applies LoRA to the QKV projection layers corresponding to text tokens. We will revise the section to highlight this intuitive explanation first.
>
> - **(C) Robustness Against Prompt Variations**
>
>   The token-zeroing baseline illustrated in ``Figure 3`` masks attention columns at inference time, which easily breaks under typos or prompt variations. Our method explicitly solves this by **fine-tuning the model weights** within our safe subspace. By optimizing the fundamental parameters via our Lagrangian-Guided algorithm, the concept erasure is deeply **embedded into the model's memory**, making it highly robust to prompt variations and adversarial attacks. This strong defense is quantitatively proven in ``Table 4``, where our method maintains a significantly lower Attack Success Rate compared to baselines.
>
>
>
> References:
>
> [1] EraseAnything: Enabling Concept Erasure in Rectified Flow Transformers. ICML 2025
>
> [2] MACE: Mass Concept Erasure in Diffusion Models. CVPR 2024
>
> [3] Minimalist Concept Erasure in Generative Models. ICML 2025

---

> > ### Author Rebuttal · Reviewer_z3eS · 2026-04-03
> >
> > Thanks for the rebuttal. I will consider changing my score after the final discussion.

---

> > > ### Author Response · Authors · 2026-04-03
> > >
> > > Thank you for your feedback! Could you please let us know if there are any remaining concerns? We'd be glad to further discuss and address them.

---

### Official Review · Reviewer_ntCY · 2026-03-12

**Soundness:** 3
**Presentation:** 3
**Significance:** 3
**Originality:** 3
**Overall Recommendation:** 5
**Confidence:** 4

**Summary:**

The paper targets concept erasure for single-stream models and a Lagrangian-based constraint optimization problem to erase undesired concepts and retain irrelevant ones. Overall, the paper demonstrates strong erasure performance compared to existing baselines.

**Compliance With Llm Reviewing Policy:**

Affirmed.

**Key Questions For Authors:**

1. What is the additional training time or memory overhead introduced by the Lagrangian-guided modulation compared to standard LoRA fine-tuning?
2. How sensitive is the final model quality to the choice of $\epsilon$? If we want to erase a novel concept, how fast can one determine an optimal $epsilon$?

**Limitations:**

See Weaknesses

**Strengths And Weaknesses:**

**Strengths:**
1. The proposed method is technically sound and outperforms previous baselines.
2. The paper is well written.

**Weaknesses:**
1. Limited Scope of Architectures: The "single-stream" definition is relatively specific for an ICML submission. Can the authors show some generalization of the proposed method to "double-stream" architectures like FLUX?

---

> ### Author Rebuttal · Authors · 2026-03-28
>
> Thank you for your constructive feedback and insightful questions! We address them as follows:
>
> - **(A) Generalization to Double-Stream Architectures (e.g., FLUX)**
>
>   Thanks! Please see our results on ``FLUX.1 [DEV]`` here: https://imgur.com/a/XjLDISo. FLUX's dual-stream blocks are **inherently decoupled**. Therefore, we only need to **fine-tune its text branch** without requiring our Stream Disentangled Framework. However, our core optimization algorithm (Lagrangian-Guided Adaptive Erasure Modulation) is completely **architecture-agnostic**. It can be directly applied to FLUX.
>
> - **(B) Training Time and Memory Overhead**
>
>   The overhead is minimal and highly manageable:
>
>   - **Time:** As detailed in ``Appendix H``, on a single H20 GPU, standard LoRA takes ``~20 minutes`` per concept, while Z-Erase takes ``~35 minutes``. This is a highly worthwhile trade-off for quality.
>   - **Memory:** The memory overhead is virtually zero. By using a first-order Taylor approximation (`Eq. 14`), we estimate the constraint via simple scalar loss differences ($\mathcal{L}\_{pr}(\theta\_{t-1}) - \mathcal{L}\_{pr}(\theta\_t)$). This completely avoids the massive memory burden of computing and storing a second full gradient graph for $\nabla \mathcal{L}\_{pr}$.
>
> - **(C) Sensitivity to Epsilon and Optimal Selection**
>
>   Model quality is highly robust to $\varepsilon$ across a wide, moderate range ($10^{-3}$ to $10^{-1}$), no need for exhaustive tuning. For a novel concept, simply look up the recommended optimal $\varepsilon$ directly by category in ``Appendix Table 6``.

---

> > ### Author Rebuttal · Reviewer_ntCY · 2026-04-03
> >
> > Thank you for all the clarifications. The rebuttal addresses most of my concerns. I will maintain my rating - Accept (5)

---

### Official Review · Reviewer_hLoU · 2026-03-13

**Soundness:** 3
**Presentation:** 3
**Significance:** 3
**Originality:** 4
**Overall Recommendation:** 5
**Confidence:** 3

**Summary:**

Z-Erase is introduced as a method with two main parts: a framework called Stream Disentangled Concept Erasure, which focuses on decoupling modalities at the parameter level, and an adaptive modulation algorithm guided by Lagrangian multipliers. The latter is particularly interesting as it dynamically balances the trade-off between removing the target concept and keeping the model’s general performance intact.

**Compliance With Llm Reviewing Policy:**

Affirmed.

**Final Justification:**

Thanks to the authors for the clear rebuttal. I am raising my score from 4 to 5. The rebuttal resolved my main concerns and, more importantly, demonstrated the universality and transferability of the proposed method

**Key Questions For Authors:**

Questions
1. The method shows resistance to basic jailbreaks (misspellings, etc.), but the paper doesn’t fully explain why. What is the underlying reason that freezing specific tokens provides this semantic robustness compared to standard fine-tuning approaches?
2. In several qualitative cases, the preservation of non-target concepts seems slightly worse than what might be achieved by precise editing models. Have you considered whether a purely data-driven approach (e.g., fine-tuning on a curated "clean" dataset) might outperform this complex optimization design? Is the Lagrangian complexity strictly necessary compared to simpler editing baselines?

**Limitations:**

yes

**Strengths And Weaknesses:**

Strengths
1. The logic behind the optimization subspace definition is straightforward but yields strong results. Specifically, explicitly freezing general image tokens effectively decouples modalities in single-stream networks, which solves the visual quality issues seen in the baselines.
2. The use of the Lagrangian method to handle the "fighting" between erasure and preservation losses is a highlight. This is a meaningful step up from static linear scalarization, providing a flexible mechanism to balance objectives during training.
3. The stability of the method is well-justified. I appreciate that the paper includes both extensive experimental validation and theoretical convergence proofs to demonstrate the robustness of the framework.

Weaknesses
1.  The name “Z-Erase” risks implying a dependency on “Z-Image,” whereas the method is actually generalizable to single-stream transformers. The authors should consider if this branding limits audience interest.
2.  I am not fully convinced effectively separating the subspace is possible just by token indexing. As depth increases, $H_{img}$ causes text and image features to mix. Consequently, $H_{img} $ in later layers will contain text information, meaning the proposed subspace might not be as “safe” or disentangled as claimed.

---

> ### Author Rebuttal · Authors · 2026-03-28
>
> Thank you for your interest in our work, especially highlighting our Lagrangian method and robust theoretical proofs! We address the questions as follows:
>
> - **(A) Naming and Generalizability**
>
>   Thank you for this valuable reminder. We completely agree that the method is generalizable to any single-stream architecture. We will consider revising the branding or adding a prominent clarification to avoid limiting audience interest.
> - **(B) Subspace** **Safety** **vs. Attention Mixing**
>
>   You raise an excellent point about semantic mixing in deeper layers. Actually, our "safe subspace" does not prevent feature mixing; rather, it isolates the parameter updates. In single-stream models, visual ($H_{img}$) and textual ($H_{txt}$) hidden states share projection weights. We restrict the LoRA updates ($\Delta W$) strictly to the textual pathway ($H_{txt}$) while completely freezing the visual backbone ($H_{img}$) from receiving gradient updates. ``Appendix Figure 15`` empirically proves this: standard fine-tuning on shared projections instantly collapses the image, but isolating the updates to $H_{txt}$ preserves visual quality perfectly.
> - **(C) Robustness to Jailbreaks (Fine-tuning vs. Token Interventions)**
>
>   To clarify, we do not freeze or zero-out tokens at inference. In fact, we explicitly demonstrate in Section 3 that simple inference-time "token-zeroing" is highly brittle against typos or jailbreaks.
>
>   Because of this brittleness, our approach relies on adversarial fine-tuning (LoRA). By directly intervening in the flow-matching velocity field ($\mathcal{L}_{erase}$) during training, the target concept is structurally and permanently suppressed from the model's latent space. This parameter-level optimization is exactly what provides the strong semantic robustness in ``Table 4``.
> - **(D) Necessity of Complex Optimization vs. Simpler Baselines**
>
>   Simpler data-driven editing may force a binary trap: under-erasure or catastrophic over-erasure. The data in our experiments clearly demonstrates why our optimization is strictly necessary:
>
>   -  **EraseAnything (simpler baseline)** [1]: Preserves irrelevant concepts moderately well (Entity $ACC_{ir}$ ``27.6%``) but fails to erase the target ($ACC_{e}$ ``24.1%``).
>   -  **UCE** [2]: Erases well but destroys general image quality (e.g., FID spikes to ``52.28`` vs. ours ``27.83``).
>   -  **Ours:** Achieves the best **Pareto-stationary balance**. We effectively erase the target (Entity $ACC_{e}$ ``23.1%``) while maintaining the highest general utility (Entity $ACC_{ir}$ ``28.9%``, highest $H_{a}$ score of ``7.4``.
>
>   As you correctly summarized, our Lagrangian modulation is necessary because it explicitly frames this erasure-preservation trade-off as a dynamically controllable hyperparameter $\varepsilon$. This strictly guarantees convergence to a Pareto optimal balance, ensuring non-target concepts are faithfully preserved while the target is cleanly erased.
>
>
>
> References:
>
> [1] EraseAnything: Enabling Concept Erasure in Rectified Flow Transformers. ICML 2025
>
> [2] Unified Concept Editing in Diffusion Models. WACV 2024

---

> > ### Author Rebuttal · Reviewer_hLoU · 2026-04-03
> >
> > thanks. I would maintain my score.

---

### Decision · Program_Chairs · 2026-04-30

**Decision:**

Accept (regular)

**Comment:**

Z-Erase is the first concept erasure method for single-stream text-to-image diffusion transformers, preventing generation collapse while achieving state-of-the-art erasure–preservation trade-offs through stream-disentangled updates and Lagrangian-guided adaptive modulation.

The rebuttal addressed most of the reviewers’ concerns, and the paper was ultimately accepted unanimously. Congratulations.